# A novel α-mangostin derivative synergistic to antibiotics against MRSA with unique mechanisms

Rile Ge,[1,2] Haiyan Zhao,[1] Qun Tang,[2] Kasemsiri Chandarajoti,[3,4] Han Bai,[5] Xiaoyang Wang,[2,6] Keyu Zhang,[2,6] Wenchong Ye,[2,6] Xiangan Han,[2] Chunmei Wang,[2,6] Wen Zhou[2,6]

**ABSTRACT** Methicillin-resistant *Staphylococcus aureus* (MRSA) remains a leading cause of hospital-acquired infections, often linked to complicated treatments, increased mortality risk, and significant cost burdens. Several antibacterial agents have been developed to address MRSA resistance. In this study, potential agents to combat MRSA resistance were explored, with the antibacterial activity of synthesized α-mangostin (α-MG) derivatives being evaluated alongside investigations into their cellular mechanisms against MRSA2. α-MG-4, featuring an allyl group at C3 of the lead compound α-MG, restored the sensitivity of MRSA2 to penicillin, enrofloxacin, and gentamicin, while also demonstrating improved safety profiles. Although α-MG-4 alone was ineffective against MRSA2, it exhibited an optimal synergistic ratio *in vitro* when combined with these antibiotics. This significant synergistic antibacterial effect was further confirmed *in vivo* using a mouse skin abscess model. Additionally, the synergistic mechanisms revealed that α-MG-4 was associated with changes in membrane permeability and inhibition of the *MepA* and *NorA* genes, which encode the efflux pumps of MRSA2. α-MG-4 also inhibited PBP2a expression, potentially by occupying a crucial binding site in a dose-dependent manner.

**IMPORTANCE** Methicillin-resistant *Staphylococcus aureus* (MRSA)'s resistance to multiple antibiotics poses significant health and safety concerns. A novel α-mangostin (α-MG) derivative, α-MG-4, was first identified as a xanthone-based PBP2a inhibitor that reverses MRSA2 resistance to penicillin. The synergistic antibacterial effects of α-MG-4 were linked to increased cell membrane permeability and the inhibition of genes involved in efflux pump function.

**KEYWORDS** α-MG derivatives, antibiotics, MRSA, synergistic antibacterial effects, *in vitro*, *in vivo*

*S*taphylococcus aureus (*S. aureus*) is a zoonotic pathogen. It produces various toxins like enterotoxin (1), which damage host cells and tissues, causing diseases such as endocarditis, bacteremia, osteomyelitis, and necrotizing fasciitis (2, 3). Antibiotic treatment is commonly used to treat *S. aureus* infection. However, irrational use of antibiotics leads to the increasing of antibiotic-resistant bacteria, especially methicillin-resistant *Staphylococcus aureus* (MRSA) (4). MRSA is resistant to numerous antibiotics causing a life-threatening condition (5, 6). The World Health Organization (WHO) considers MRSA as a "high-priority" pathogen (7). Therefore, developing novel strategies to treat MRSA infection is urgent.

Development of new treatment for bacterial infections such as utilizing drug combination regimen presents an attractive approach for managing resistant bacterial infection. Drug combination not only saves time, and it also reduces the doses and cost of treatments (6). A variety of secondary metabolites from plants alleviate

**Peer Reviewers** Erdal Özbek, Diyarbakır Gynecology and pediatric hospital, Diyarbakır, Bağlar, Turkey; Hafidha Salim AL-Hattali, Sultan Qaboos University College of Medicine and Health Science, ALSeeb, Muscat, Oman

Address correspondence to Chunmei Wang, wangchunmei@shvri.ac.cn, or Wen Zhou, zhouwen60@126.com.

Rile Ge and Haiyan Zhao contributed equally to this article. The author was determined by their conribution to the article.

The authors declare no conflict of interest.

See the funding table on p. 21.

pathogenic attacks including baicalin (8, 9), quercetin (10), epicatechin (11), and ginkgo biloba flavonoids (12) and demonstrate good antibacterial synergistic effects. α-Mangostin (α-MG), a xanthone compound isolated from *Garcinia cambogia*, exhibits multiple biological activities, including anti-inflammatory (13–17), antibacterial (18), antioxidative (19), and anti-cancer properties (20). However, its clinical application is hindered by low water solubility and hemolytic toxicity. Structural modifications to α-MG resulted in several xanthone derivatives with potential antibacterial activity and toxicity reduction, suggesting the importance of a core pharmacophore of α-MG and its structure-activity relationship (21). Our previous study indicated that modifying α-MG structure with an acetyl group at C1 decreased its toxicity and enhanced anti-bacterial efficacy by disrupting the bacterial membrane (22). However, few xanthone-derived compounds have been designed and explored as anti-bacterial agents in combination with the existing antibiotics to combat drug-resistant bacteria.

In this study, various structures of α-mangostin (α-MG) derivatives were synthesized and screened against Gram-positive and Gram-negative bacteria. The findings demonstrated that α-MG substituted with an allyl group at C3, forming α-MG-4, reduced toxicity to A549 and L0-2 cells and lowered hemolysis. These α-MG derivatives enhanced antibacterial activity by disrupting the bacterial membrane. Additionally, selected xanthone-derived compounds were evaluated as anti-MRSA2 agents in combination with various classes of existing antibiotics. The α-MG-4 derivative was identified as a synergist, showing synergistic effects with penicillin, enrofloxacin, and gentamicin against MRSA2. The cellular mechanisms behind these synergistic effects were further explored, focusing on bacterial membrane integrity and efflux pump inhibition.

## RESULTS

### The *in vitro* synergistic effects of α-MG derivatives and antibiotics against MRSA2

Eleven α-MG derivatives described in Fig. 1 were synthesized as previously reported (22), We firstly evaluated the MICs of the synthesized α-MG derivatives against *S. aureus* (a representative of Gram-positive bacterium) and *Escherichia coli* (a representative of Gram-negative bacterium) and reported (Table 1). The synergistic inhibitory effects of α-MG derivatives and various group of MRSA2 antibiotics were described in Table 2. We performed the screening by determining the fractional inhibitory concentration index (FICI) to further select the appropriate antibiotics to demonstrate the synergistic effects. Beta-lactam, fluoroquinolone, and aminoglycoside were selected, including penicillin, enrofloxacin, and gentamicin, respectively, for further study. When the FIC value is less than 0.5, the compound is considered as a synergistic agent. Therefore, α-MG-2, α-MG-3, and α-MG-4 were selected for further synergistic investigation (Fig. 2).

### *In vitro* hemolytic activity and cytotoxicity of α-MG derivatives

To identify effective α-MG-derived synergists with favorable safety profiles, we assessed their hemolytic activity and cytotoxicity. Hemolysis of rabbit red blood cells (RBC) was determined against the increasing concentration (3.125–100 mg/L) of the tested compounds. α-MG and α-MG-2 showed similar cell lytic activity while α-MG-3 and α-MG-4 showed lesser extent in RBC lysis (Fig. 3A). Cell viability determination of the α-MG derivatives against A549 (lung cells) and L0-2 (hepatocyte cells) showed that α-MG, α-MG-2, and α-MG-3 exhibited the decrease in cell viability at a concentration of 10–20 mg/L. However, α-MG-4 demonstrated more than 60% of cell viability at 40 mg/L (Fig. 3B). Similarly, the determination obtained on L0-2 cells also showed that α-MG-2 is markedly toxic to L0-2 cells compared with that of α-MG. However, α-MG-4 displayed a safety profile against L0-2 cells showing cell viability of more than 60% at 40 mg/L (Fig. 3C). These findings were consistent with previous reports on hemolytic activity and cytotoxicity of α-MG, suggesting that α-MG derivative caused certain toxicity to the cells due to its chemical structure (22, 23). The results indicated that structural modifications

| NO. | X | Y | Z | W | $R_1$ | $R_2$ | $R_3$ |
|---|---|---|---|---|---|---|---|
| α-MG | O | O | O | O | H | H | H |
| α-MG-1 | O | O | O | H | H | H | H |
| α-MG-2 | OCO | O | OCO | O | $CH_3$ | H | $CH_3$ |
| α-MG-3 | O | O | O | O | Bn | Bn | Bn |
| α-MG-4 | O | O | O | O | H | $CH_2CH=CH_2$ | H |
| α-MG-5 | O | OCO | OCO | O | H | $CH_3$ | $CH_3$ |
| α-MG-6 | O | O | OCO | O | H | H | $CH_3$ |
| α-MG-7 | O | O | O | O | $COCH(CH_3)_2$ | $COCH(CH_3)_2$ | $COCH(CH_3)_2$ |
| α-MG-8 | O | O | O | O | $COCH(CH_3)_2$ | H | $COCH(CH_3)_2$ |
| α-MG-9 | O | O | O | O | H | $(CH_2)_5OH$ | $(CH_2)_5OH$ |
| α-MG-10 | O | O | O | O | $COCH(CH_3)_2$ | $CH_2CH=CH_2$ | $CH_2CH=CH_2$ |
| α-MG-11 | O | O | O | O | H | $CH_2CH=CH_2$ | $(CH_2)_5OCOCH_3$ |

**FIG 1** Structures of a-MG derivatives.

played a role in decreasing toxicity of α-MG derivatives. α-MG-4 exhibited favorable safety profiles and was selected for further investigation on the synergistic effects against MRSA2 in combination with antibiotics.

### α-MG-4 in combination with antibiotics demonstrated synergistic effects against MRSA2 using a checkerboard assay

We proceeded to assess the synergistic effect of α-MG (Fig. 4A) and α-MG-4 (Fig. 4B) in combination with three selected antibiotics on MRSA2. On the heat map diagram, darker blue indicates the bacterial growth. FICI less than 0.5 indicates a synergistic effect while FICI greater than 0.5 indicates an additive effect or indifferent effect. The FICI of α-MG-4 in combination with penicillin, enrofloxacin, and gentamicin were determined to be 0.015, 0.017, and 0.018, respectively, suggesting the synergistic effect of α-MG-4. In contrast, α-MG in combination with antibiotics showed FICI more than 0.5 and darker blue expanded in a larger area.

### Time-killing assay revealed synergistic effects of α-MG-4 in combination with antibiotics

Time growth curve (Fig. 5A) and time-killing curve (Fig. 5B) assays were conducted to determine the synergistic effect of α-MG-4. The antibacterial activity against MRSA2 of α-MG-4 was exhibited within 2 h as observed by the decrease in optical density (O.D.) and the decline in CFU, consistent with its MIC (>256 mg/L). MRSA2 demonstrated

**TABLE 1** MIC (mg/L) of α-mangostin and its derivatives against different bacteria strains[a]

| Number | Gram positive (*S. aureus*) | | Gram negative (*E. coli*) | |
|---|---|---|---|---|
| | ATCC 29213 | MRSA2 | ATCC 25922 | CRE-1 |
| α-MG | 0.5–2 | 0.5–2 | >256 | >256 |
| α-MG-1 | 16 | 16 | >256 | >256 |
| α-MG-2 | >256 | >256 | >256 | >256 |
| α-MG-3 | 128 | >256 | >256 | >256 |
| α-MG-4 | >256 | >256 | >256 | >256 |
| α-MG-5 | >256 | >256 | >256 | >256 |
| α-MG-6 | 4 | 2–4 | >256 | >256 |
| α-MG-7 | >256 | >256 | >256 | >256 |
| α-MG-8 | >256 | >256 | >256 | >256 |
| α-MG-9 | >256 | >256 | >256 | >256 |
| α-MG-10 | >256 | >256 | >256 | >256 |
| α-MG-11 | >256 | >256 | >256 | >256 |
| Penicillin | 0.5 | 32 | 32 | >128 |
| Enrofloxacin | 0.125 | 32 | <0.03125 | 8 |
| Gentamicin | 0.25–0.5 | 32 | 1–2 | >128 |
| Rifampicin | 0.015 | 128 | / | / |
| Tetracycline | 0.25 | 16–32 | 1 | 64 |
| Clindamycin | 0.25 | 256 | / | / |
| Azithromycin | 0.125 | 32 | 2 | 64 |
| Polymyxin | / | / | 2 | 1–4 |
| Vancomycin | 1 | 1 | / | / |

[a]ATCC 29213 and ATCC 25922 are standard sensitive strains, and MRSA2 and CRE-1 are clinically isolated in our laboratory. MIC ≤ 4 mg/L means the strain is sensitive to the antibiotics; 8 ≤ MIC ≤ 16 mg/L means the strain is sensitive to the antibiotics in a dose-dependent manner; MIC ≥ 32 mg/L means the strain is resistant to the antibiotics. "/" means no determination. Vancomycin as a control for Gram-positive bacteria, enrofloxacin as a control against Gram-negative bacteria. Ceriotti et al. "Clinical and Laboratory Standards Institute" (2018).

resistance to antibiotics including penicillin, enrofloxacin, and gentamicin. Interestingly, the addition of α-MG-4 overcame the resistance of the three antibiotics and demonstrated synergistic effects, resulting in an increase in bactericidal activity. The optimal inhibitory concentrations of α-MG-4 in combination with penicillin, enrofloxacin, and gentamicin were determined to be 8 mg/L + 1 mg/L, 8 mg/+0.3125 mg/L, and 16 mg/L + 0.3125 mg/L, respectively. These concentrations were preferentially utilized for further exploration in the subsequent assays.

**TABLE 2** FICI of α-mangostin derivatives with antibiotics against MRSA2[a]

| | MRSA2 | | | | | | |
|---|---|---|---|---|---|---|---|
| Number | Penicillin (β-lactams) | Enrofloxacin (quinolones) | Gentamicin (aminoglycosides) | Azithromycin (macrocyclic lipids) | Rifampicin (rifamycin) | Tetracycline (tetracyclines) | Clindamycin (lincosamide) |
| α-MG | 0.75 | 0.51 | 0.5625 | 2 | 0.52 | 0.51 | 2 |
| α-MG-1 | 0.5 | 1 | 0.375 | 0.51 | 0.51 | 2 | 2 |
| α-MG-2 | 0.019 | 0.01 | 0.01 | 1 | 0.51 | 0.056 | 2 |
| α-MG-3 | 0.023 | 0.02 | 0.017 | 1 | 0.51 | 0.093 | 2 |
| α-MG-4 | 0.015 | 0.035 | 0.017 | 2 | 2 | 0.51 | 2 |
| α-MG-5 | 1 | 1 | 1 | 1 | 2 | 2 | 2 |
| α-MG-6 | 0.375 | 0.51 | 0.5 | 0.51 | 2 | 0.26 | 2 |
| α-MG-7 | 1 | 1 | 1 | 2 | 2 | 2 | 2 |
| α-MG-8 | 0.75 | 1 | 0.51 | 1 | 2 | 2 | 2 |
| α-MG-9 | 0.5 | 1 | 0.53 | 1 | 2 | 2 | 2 |
| α-MG-10 | 1 | 1 | 1 | 1 | 2 | 2 | 2 |
| α-MG-11 | 1 | 1 | 1 | 1 | 2 | 2 | 2 |

[a]When the FICI ≤0.5, it was synergistic, 0.5 < FICI ≤ 1 was additive, 1 < FICI ≤ 2 was irrelevant, and FICI >2 is antagonistic. The evaluation standard was in line with reference 36.

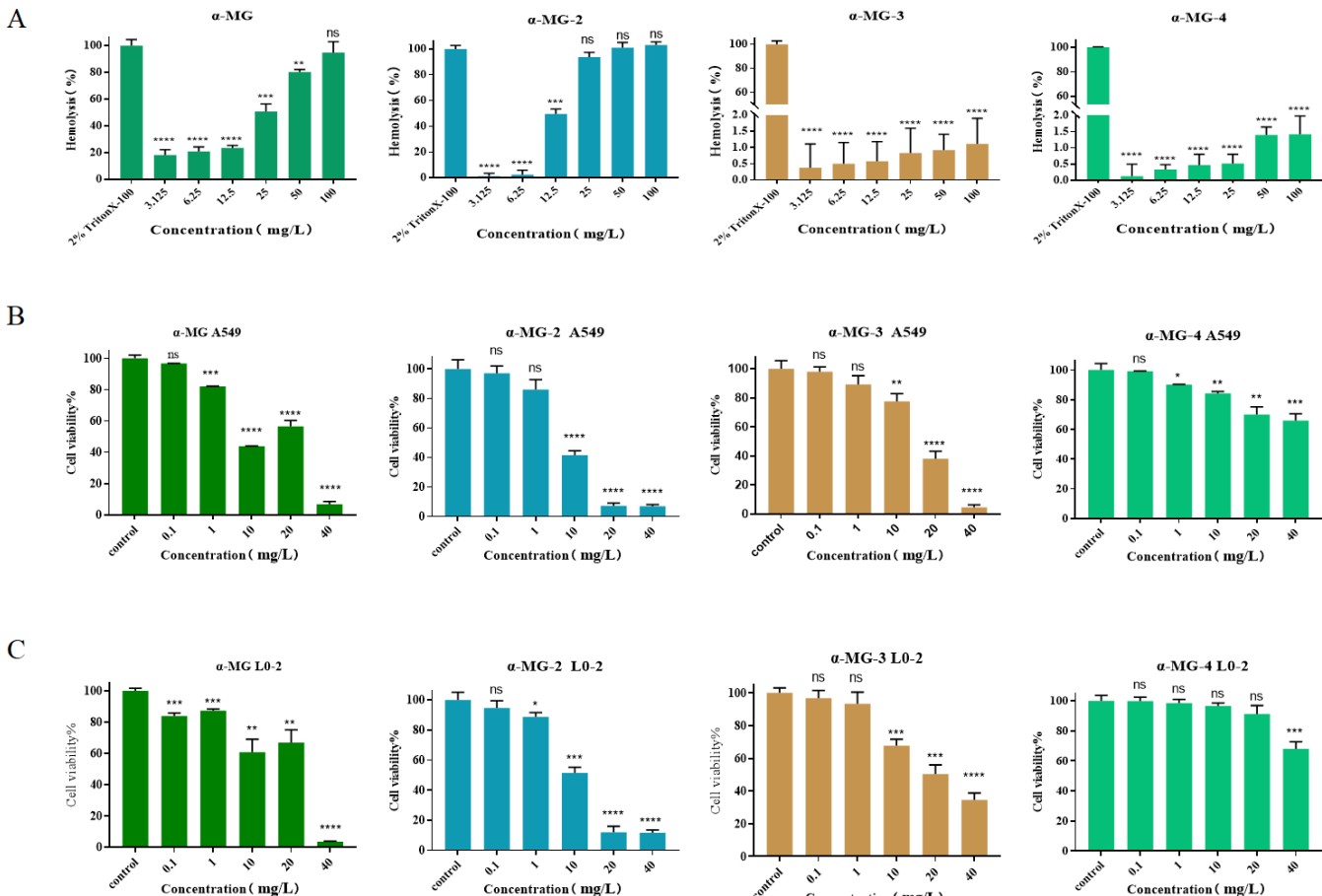

**FIG 2**  Chemical structure of a-MG-2, a-MG-3, and a-MG-4.

## *In vivo* therapeutic effects of α-MG-4 in combination with antibiotics

To evaluate therapeutic effects of α-MG-4 in combination with antibiotics against MRSA, we established a murine skin abscess model infected with MRSA2. Penicillin and vancomycin were antibiotics selected in this *in vivo* study. We analyzed the corresponding CFU density resided at the infected area (Fig. 6A), inflammatory mediator, interleukin 6 (IL-6) (Fig. 6B), and skin histopathology. Penicillin was a tested antibiotic while vancomycin was a positive control. On day 2, administration of either α-MG-4 or penicillin alone was unable to resolve the abscess. However, co-administration of α-MG-4

**FIG 3**  Hemolytic activity and cytotoxicity of a-MG derivatives. (A) Hemolytic activity of a-MG, a-MG-2, a-MG-3, and a-MG-4 at different concentrations. (B) Cell viability (%) of A549 cells after incubation with different concentrations of a-MG, a-MG-2, a-MG-3, and u-MG-4. (C) Cell viability (%) of 10–2 cells after incubation with different concentrations of a-MG, a-MG-2, a-MG-3, and a-MG-4. All data were representative of three independent experiments and were presented as the mean ± SD. Statistical significance was indicated with *, *P* value, * <0.05 , **<0.01, ***<0.001, and ****<0.0001. ns indicated results were not significant.

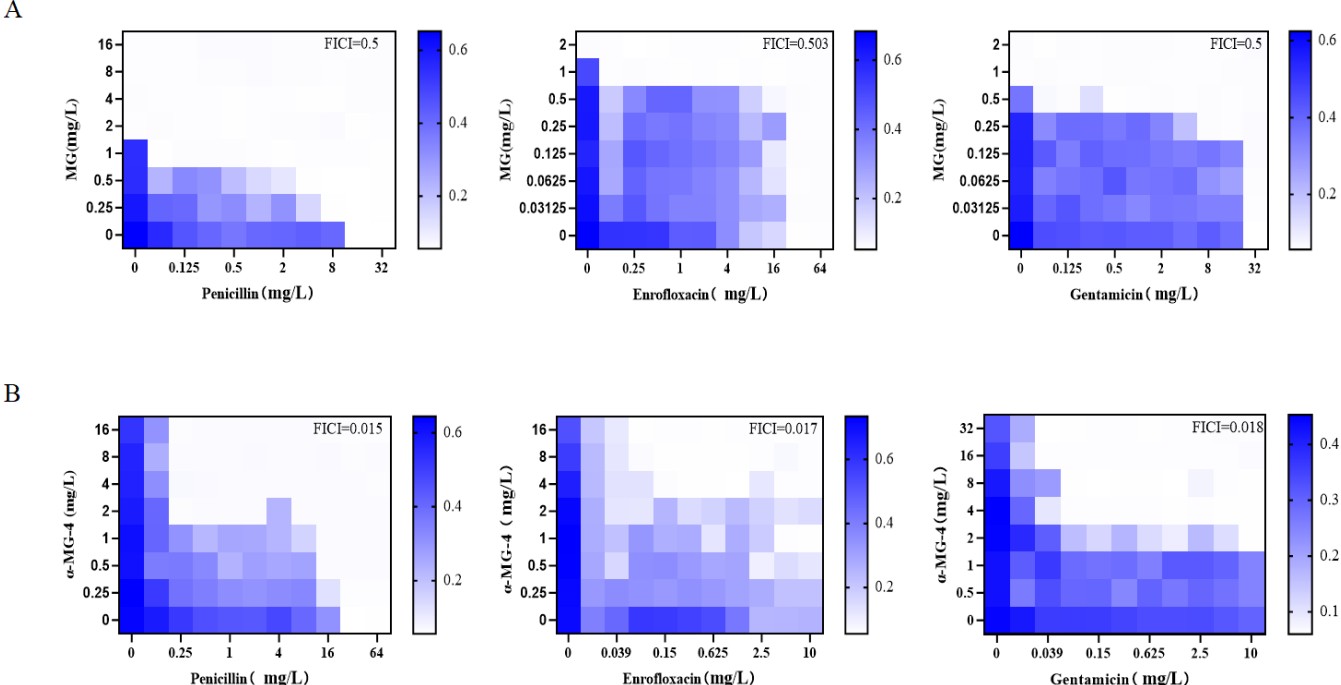

**FIG 4** Heat maps of the synergistic, additive, or indifferent effects between a-MG derivatives and antibiotics with a checkerboard assay. (A) Heat maps of MRSA2 exposed to a-MG and three antibiotics. (B) Heat maps of MRSA2 exposed to a-MG-4 and three antibiotics. Darker blue indicated more bacterial growth. FICI less than 0.5 indicated synergy, while FICI greater than 0.5 indicated additive effects or indifferent effects.

and penicillin resulted in a drastic reduction in abscess size (Fig. 6C), accompanied by a significant decrease in CFU number and concentration of IL-6 ($P \leq 0.01$–0.001). Histological analysis via H&E staining revealed the reduced inflammatory mediator infiltration to the cells and intact tissue structure in specimens when treated with the co-administered α-MG-4 and penicillin (Fig. 6D). Nevertheless, in comparison to vancomycin for anti-MRSA2 activity, a therapeutic effect of the α-MG-4 combination remained relatively inferior. This observation was supported by residual bacteria observed in the histological sections (Fig. 6A).

## α-MG-4 in combination with antibiotics lysed MRSA2 cell membrane

A synergistic effect of α-MG-4 with or without antibiotics was observed to investigate synergistically the action mechanisms against MRSA2. Scanning electron microscopy (SEM) and transmission electron microscopy (TEM) images of MRSA2 treated with either individual α-MG-4 (8 mg/mL) or penicillin (1 mg/L) or enrofloxacin (0.3125 mg/L) or gentamicin (0.3125 mg/L) as well as their combinations resulted in distinct MRSA2 surfaces (Fig. 7). Untreated MRSA2 exhibited a complete morphology, clear cell membrane boundary, and a smooth surface. Treatment with the three antibiotics failed to disrupt the MRSA2 surface morphology. However, when MSRA2 is exposed to the treatment containing both antibiotics and α-MG-4, SEM and TEM images showed slightly leakage of cellular contents at the bacterial surface. When co-administered, the synergistic effect caused α-MG-4 attachment to the bacterial surface (red arrow) and changes in bacterial morphology, resulting in the increase in bacterial content leakage as observed as the blurry boundary of the membrane. The observations from TEM were highly consistent with that obtained from the SEM.

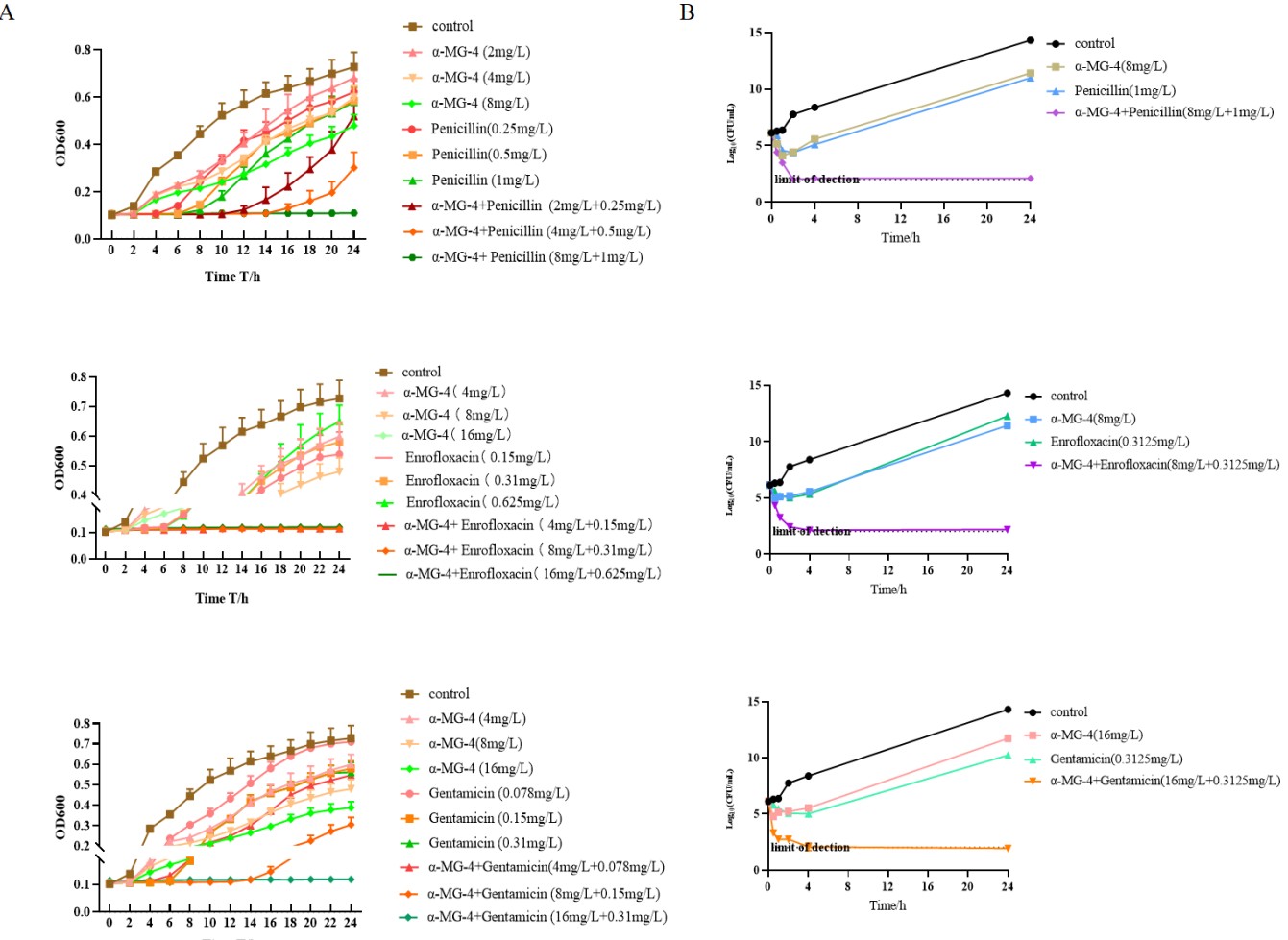

**FIG 5** Time-killing assays of MRSA2 exposed to a-MG-4 and different antibiotics. (A) The growth curve of MRSA2 exposed to a-MG-4 and three antibiotics. (B) The time-killing assays of MRSA2 exposed to o-MG-4 and three antibiotics. All data are representative of three independent experiments and are presented as the mean SD.

## Live/dead cell staining analysis indicated synergistic effects of α-MG-4 in combination with antibiotics

The antibacterial effects of the combination between α-MG-4 and antibiotics were further confirmed through the live/dead bacteria staining using fluorescent agents (Fig. 8). DAPI labels live cells while YO-PRO-1 labels dead cells. Melittin, a 26-amino acid peptide, which is a honey bee venom, served as a positive control (Fig. 8A). Upon treatment with α-MG-4, green fluorescent was observed. Moreover, when α-MG-4 was combined with penicillin or enrofloxacin or gentamicin to treat MRSA2, the green fluorescent intensity markedly increased (Fig. 8B through D). The inhibitory effects on MRSA2 were measured and indicated that combining α-MG-4 and the antibiotics affected MRSA2 equivalent to that of melittin (Fig. 8E).

## α-MG-4 displayed efflux pump inhibition

To further investigate whether disrupting of the bacterial cell membrane was associated with the efflux pump, the accumulation of ethidium bromide (EtBr) was analyzed. Enrofloxacin was selected in this study due to fluoroquinolones influencing the function of the efflux pump. In a blank group, the accumulation of EtBr reached a certain extent, achieving a balance by exporting out to the extracellular environment due to the

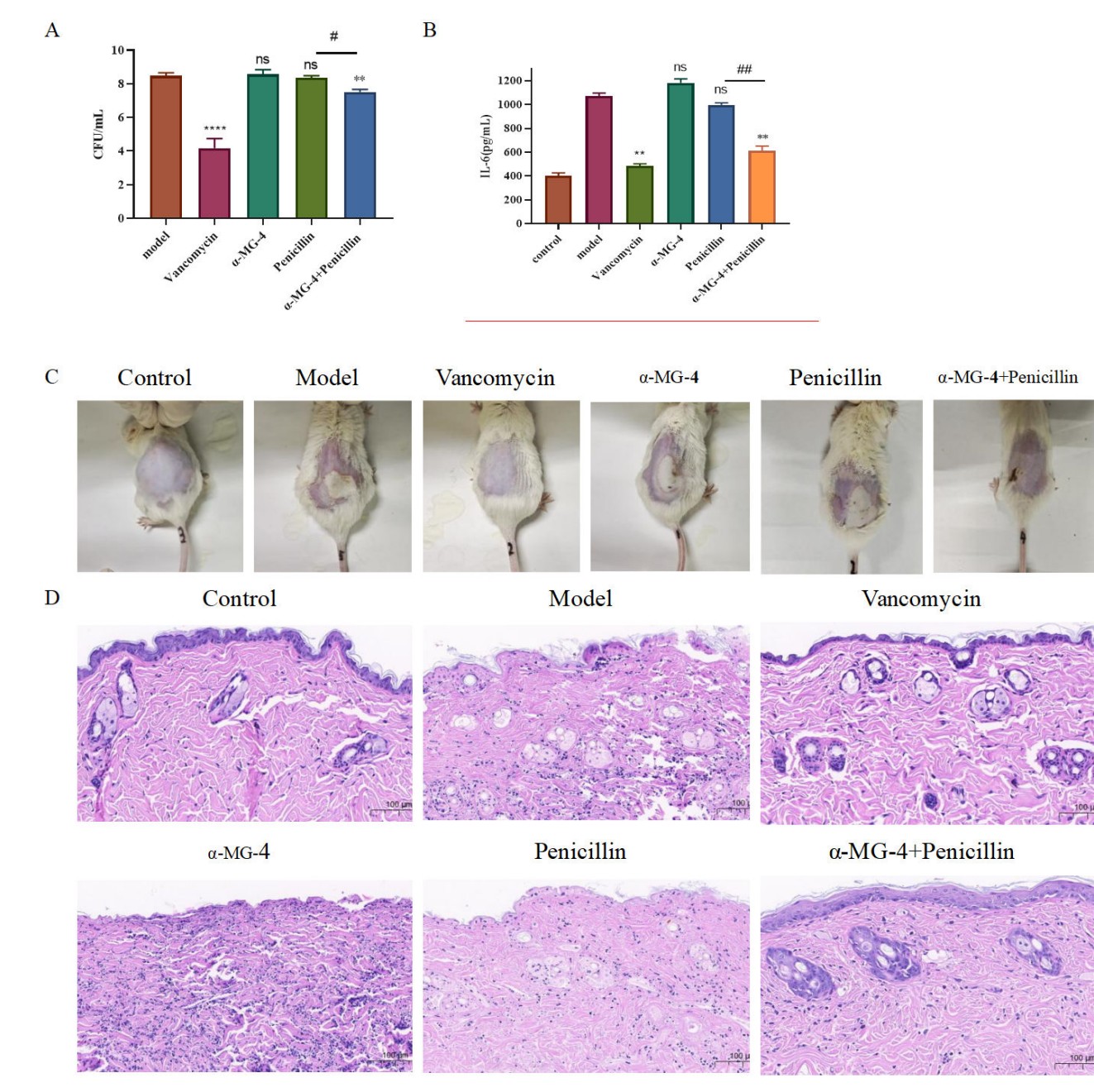

**FIG 6** A synergistic effect of a-MG-4 plus penicillin in a MRSA2-infected mouse skin abscesses model. MRSA2-infected mice were treated with vancomycin, penicillin, or a-MG-4 in combination with penicillin. DMSO was served as the blank control. (A) The bacterial burden at the abscess site in mice. Statistical significance was indicated with *or #. $P$ value, *and # < 0.05, ** and ## ≤ 0.01, *** and ###< 0.001, and **** and #### < 0.0001. ns indicates results were not significant. (B) The production of the inflammatory mediator IL-6 in the abscess was detected using enzyme-linked immunosorbent assays (ELISA). (C) Restriction of the abscess area in the infected model. Each image showed dermonecrotic lesions from a representative mouse for each group. (D) Abscess histopathology in the infection model. H&E staining revealed the location of abscesses. All data were presented as the mean ± SD.

absence of an efflux pump inhibitor (Fig. 9A). An increased accumulation of EtBr caused by α-MG-4 was observed. However, an efflux pump inhibitor, carbonyl cyanide *m*-chlorophenylhydrazone (CCCP), showed no effect on the accumulation of EtBr suggesting a mode of action, which differed from that of α-MG-4. The limited increase observed with CCCP possibly corresponded to its damaging effect on the cell membrane. As

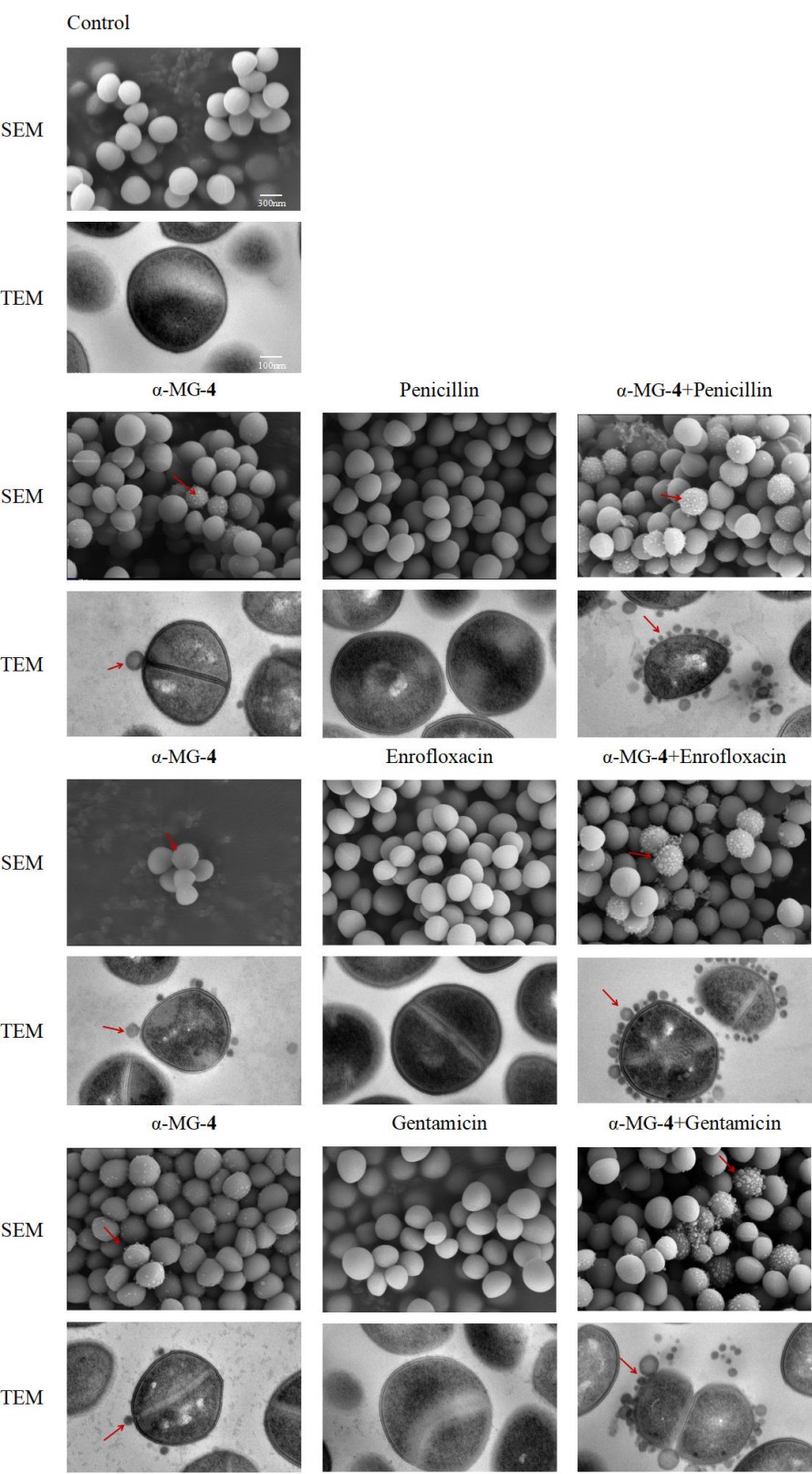

**FIG 7** SEM and TEM analyses of MRSA2 after exposure to a-MG-4 and three antibiotics. (A) MRSA2 with no treatment. (B) MRSA 2 treated with 8 mg/L a-MG-4 and 1 mg/L penicillin. (C) MRSA2 treated with 8 mg/L a-MG-4 and 0.3125 mg/L enrofloxacin. (D) MRSA2 treated with 16 mg/L a-MG-4 and 0.3125 mg/L gentamicin (SEM: scale bar, 300 nm, TEM: scale bar,100 mm).

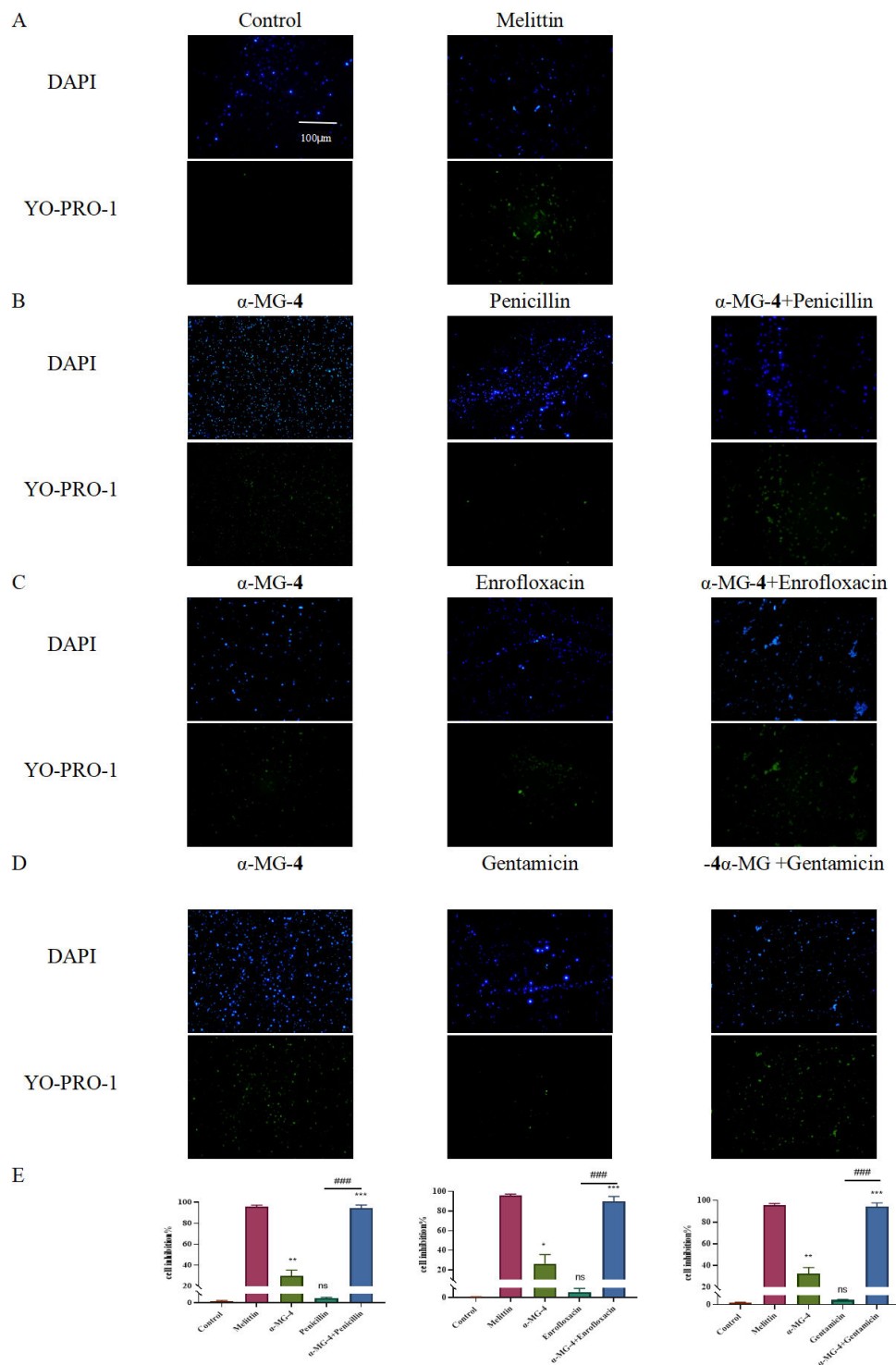

**FIG 8** Live/dead cell staining analysis of MRSA2 exposed to a-MG-4 and different antibiotics. Live bacteria were stained blue (DAPI), and dead bacteria were stained green (YO-PRO-1) (scale bar,100 µm). (A) MRSA2 with no treatment. (B) MRSA2 treated with 8 mg/L a-MG-4 and 1 mg/L penicillin. (C) MRSA2 treated with 8 mg/L a-MG-4 and 0.3125 mg/L enrofloxacin. (D) MRSA2 treated with 16 mg/L a-MG-4 and 0.3125 mg/L gentamicin. (E) Ratio of live bacteria according to live dead bacterial staining results. All data represented three independent experiments and were presented as mean standard deviations. Statistical significance was indicated with * or #. *P* value * and # < 0.05, ** and ** < 0.01, *** and ###< 0.001, and **** and #### < 0.0001. ns indicated results were not significant.

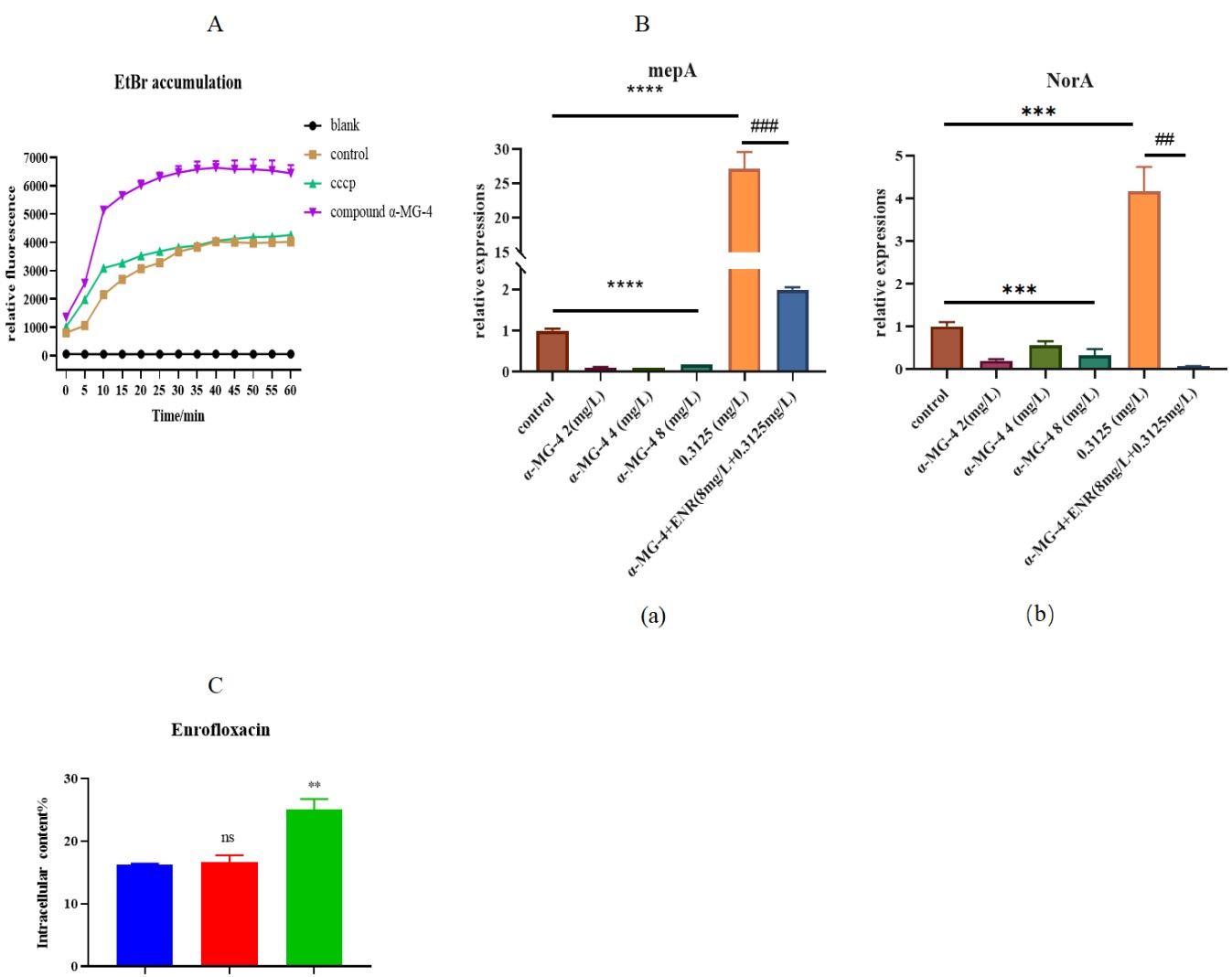

**FIG 9** Inhibition of a-MG-4 on efflux pump. (A) Accumulation of EtBr in MRSA2. In the presence of a-MG-4 (8 mg/L) and positive control CCCP (1 mg/L). CCCP, carbonyl cyanide m-chlorophenylhydrazone. (B) The effects of a-MG-4 and antibiotics on the transcription of MRSA2-related efflux pump genes (A) mepA and (B) NorA. Detection of changes in the expression of mepA and NorA by a-MG-4 (2, 4, and 8 mg/L), enrofloxacin (0.3125 mg/L), and a-MG-4 (8 mg/L) combined with enrofloxacin (0.3125 mg/L). (C) HPLC determination of intracellular enrofloxacin content in the presence of a-MG-4 and CCCP. All data represented three independent experiments and were presented as mean ± standard deviations. Statistical significance was indicated with * or #. *P* value, * and # < 0.05, ** and ## < 0.01, *** and ###, < 0.001, and **** and ####<0.0001, .ns indicated results were not significant.

anticipated, changes in the relative expression of efflux pump genes were observed. α-MG-4 inhibited *MepA* [Fig. 9B (a)] and *NorA* [Fig. 9B (b)] genes in a concentration-dependent manner. When enrofloxacin (0.3125 mg/mL) was treated alone, the relative expression of the *MepA* and *NorA* genes increased approximately 28 times and 5 times, respectively. However, the relative expression of *MepA* (###*P* < 0.001) and *NorA* (##*P* < 0.01) genes decreased significantly when treating with α-MG-4 in combination with enrofloxacin (Fig. 9B). This observation further suggested the essential presence of α-MG-4 in achieving synergistic antibacterial effects mediated by the inhibition of the *MepA* and *NorA* genes encoding the efflux pump. HPLC determination of intracellular enrofloxacin content indicated that α-MG-4 significantly retained enrofloxacin in MRSA2 (*P* < 0.01) (Fig. 9C).

## α-MG-4 exhibited penicillin-binding protein 2a inhibition

One of the mechanistic resistance traits of MRSA to penicillin is involved with penicillin-binding protein 2a (PBP2a) (23). We investigated the inhibitory effects of α-MG-4 on this particular resistant protein. At a concentration of 62.5 mg/L, the active site of PBP2a was completely occupied by α-MG-4 instead of Bocillin FL (Fig. 10A). Furthermore, PBP2a expression was inhibited in a dose-dependent manner, with the inhibition plateau achieved at 8 mg/L of α-MG-4. In determining PBP2a expression of MRSA2, a significant suppression was observed when α-MG-4 combined with penicillin ($^{###}P \ll 0.01$–$0.001$) whereas individual penicillin was unable to suppress PBP2a (Fig. 9B). A result from western blot analysis provided additional evidence to explain the synergy between

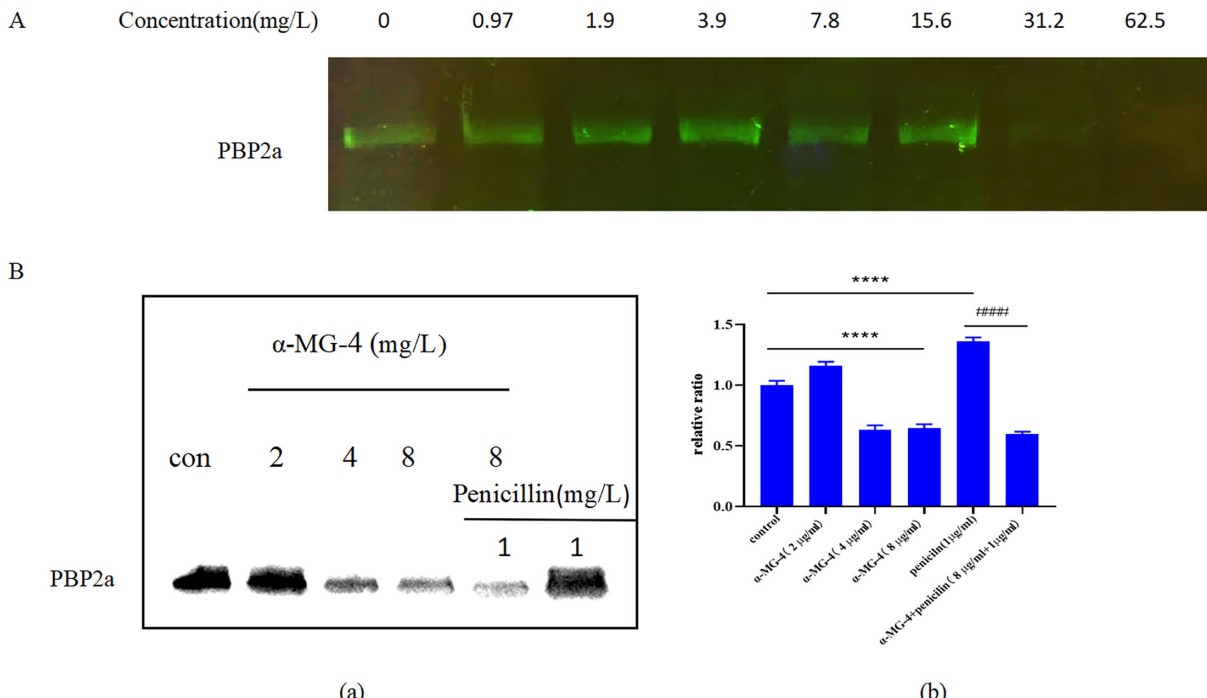

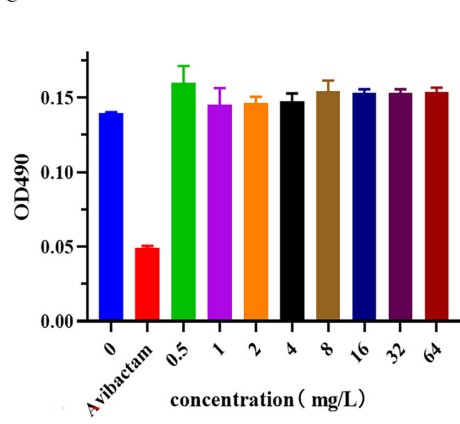

**FIG 10** Inhibition of PBP2a by a-MG-4. (A) a-MG-4 inhibited the binding of MRSA2 PBP2a and Bocillin FL reagent. (B) a-MG-4 inhibited the expression of PBP2a in MRSA2. (a) Changes in the expression of PBP2a treated by a-MG-4 (2, 4, and 8 mg/L), penicillin (1 mg/L), and a-MG-4 (8 mg/L) combined with penicillin (1 mg/L). (b) The quantitative analysis of PBP2a. (C) B-lactamase in MRSA was not inhibited by a-MG-4. Inhibition of β-lactamase activity by a-MG-4 at 0.5–64 mg/L. Avibactam was a positive control. All data represented three independent experiments and were presented as mean ± standard deviations. Statistical significance was indicated with * or #, *P* value, * and #< 0.05, ** and ##, < 0.01, *** and ###, < 0.001, and **** and ####<0.0001,. ns indicated the results were not significant.

α-MG-4 and penicillin. Additionally, β-lactamase analysis assay revealed no inhibitory effect of α-MG-4 on β-lactamase activity (Fig. 10C).

## DISCUSSION

Continuing of MRSA resistance facilitates the search for novel strategies to combat MRSA. A combination of anti-microbial agents is one of the effective approaches in tackling the resistance (24). Active components from plants containing anti-bacterial activity have been intensively studied. Compounds such as α-MG when combining with existing antibiotics showed synergistic effects against antibiotics resistance bacteria (8).

In our previous work, we synthesized and explored structural modifications on α-MG. We discovered some α-MG derivatives displaying anti-microbial activities. Among these derivatives (Table 1), α-MG-4, administered as an individual compound, lacked anti-bacterial activity; however, it showed significant synergistic effects against MRSA2 when combined with either penicillin or enrofloxacin or gentamicin. α-MG-4 restored antibacterial effectiveness of penicillin, enrofloxacin, and gentamicin against MRSA2 with FICI less than 0.035 (Table 2). Notably, safety profiles of α-MG-4 on RBC and cell lines, including lessened hemolytic activity and minimal cytotoxicity, were superior to that of α-MG and other derivatives (Fig. 3). The synergistic effects of α-MG-4 when combined with penicillin, enrofloxacin, and gentamicin against MRSA2 were further confirmed by bacterial growth and bactericidal curve assays (Fig. 5). Interestingly, α-MG-4 demonstrated reversing activity against MRSA2 resistance when co-administered with penicillin. Combination of α-MG-4 and penicillin was superior than the other two antibiotics observed *in vitro* (Fig. 4 and 5) and in mouse abscess models (Fig. 6). The antibacterial activity of α-MG-4 combined with penicillin was more potent than when these two agents individually acted. α-MG-4 significantly lowered the IL-6 level in MRSA2-infected mice. We observed that none of the discomforts resulted from the abscesses in mice. There were no physical and apparent symptoms presented related to the toxicity of α-MG-4 used in the *in vivo* study.

The resistance mechanisms of penicillin, enrofloxacin, and gentamicin differ. For β-lactam antibiotics, e.g., penicillin, MRSA typically employs two main resistance mechanisms. First, it expresses β-lactamase, which breaks the β-lactam ring, a pharmacophore responsible for the antibacterial activity. Second, it produces PBP2a, which reduces the affinity of β-lactam antibiotics to PBP2 transpeptidase. MRSA develops resistance to aminoglycosides mainly by modifying the synthesis of aminoglycoside transferases (25). Resistance to fluoroquinolones involves mutations in DNA synthase (topoisomerase II and topoisomerase IV), as well as the expression of efflux pumps resulting in the decrease of drug concentrations (26). Major Facilitator Superfamily (MFS) including *NorA*, *NorB*, and *NorC* genes and Multidrug and Toxic Compound Extrusion (MATE) family including *MepA* gene primarily control the inhibition of efflux pumps (27). Additionally, disruption at bacterial cell membrane is an essential target for inhibition of bacterial resistance (28). Thus, the reversal mechanisms of MRSA resistance to the aforementioned antibiotics can be managed differently.

α-MG-4 was capable of restoring the susceptibility of the antibiotics. We conducted cellular analysis of its synergistic mechanisms. Results from SEM and TEM indicated significant changes in the surface, morphology, and internal structure of MRSA2 after the co-treatment of α-MG-4 and antibiotics (Fig. 7). These changes were consistent with previous observation suggesting that α-helix peptides played a role in increasing membrane permeability, thereby exhibiting antimicrobial effects (29). It was hypothesized that, similar to the membrane pores created by *Alpinia purpurata* lectin (ApuL), α-MG-4 may target the cell membrane, contributing to the pore formation. Consequently, α-MG-4 exerts synergistic effects on the antibiotics. Further evaluation of membrane integrity supported the increased damaging ability of α-MG-4 to the MRSA2 membrane when combined with antibiotics (Fig. 8). The observation was similar to when the core structure of the parent, α-MG, with the xanthone structure was investigated. The lipotropic effect of compound containing xanthone may contribute to the pore

formation at the cell membrane (30, 31). Furthermore, an allyl substitution introduced of α-MG-4 may be benificial to the reduced MRSA membrane integrity due to its lipophilicity.

In addition to membrane permeability and integrity, efflux pumps are linked to antibiotic resistance. The accumulation of EtBr served as an indicator of efflux pump inhibition (32). α-MG-4 exhibited an inhibitory effect on the efflux pump, as evidenced by the increased accumulation of EtBr (Fig. 9) as well as the inhibition of *MepA* and *NorA* genes in a dose-dependent manner. *MepA* gene encodes protein of the MATE family, which facilitates an efflux of fluoroquinolones (26). *NorA* gene encodes NorA protein, an efflux pump in the MFS family that expels fluoroquinolones (33). Recovery of enrofloxacin susceptibility to MRSA2 when co-administered with α-MG-4 resulted from the inhibition of expression of the efflux pumps encoded by *MepA* and *NorA* genes. Thus, enrofloxacin was retained inside the bacteria as determined by the HPLC. For penicillin resistance, we assessed the inhibitory effects of α-MG-4 on the resistant proteins PBP2a and β-lactamase. Bocillin FL, a fluorescent reagent capable of competitively occupying the active site of PBP2a, is commonly used to assess inhibition activity against PBP enzymes (34). Interestingly, α-MG-4 at high concentration (31 and 62.5 mg/mL) caused the disappearance of Bocillin FL fluorescence, possibly suggesting that α-MG-4 may occupy the active site of PBP2. The inhibition of PBP2a by α-MG-4 occurred in a dose-dependent manner. However, β-lactamase was unaltered by α-MG-4 indicating that α-MG-4 may bind to and inhibit other domains of PBP2a due to the similar active sites for them. At a high concentration of 31 or 62.5 mg/L, the occupancy on the active site of PBP2a may be non-specific. α-MG-4 may bind to and inhibit the allosteric site located on PBP2a (35). When co-administered with α-MG-4 and penicillin, PBP2a expression was markedly decreased demonstrating the enhanced effect corresponding to anti-MRSA2 activity (Fig. 10). The possible mechanism was inhibition of PBP2a in MRSA2 by α-MG-4 synergistically sensitizing the antibacterial activity of penicillin. It was worth noting that α-MG-4 was the first reported PBP2a inhibitor with a xanthone structure, introducing a novel class of scaffolds for combating β-lactam antibiotic resistance.

In summary, α-MG-4, containing an allyl group substituted to the xanthone scaffold, exhibited a notable synergistic effect in enhancing the sensitivity of penicillin, enrofloxacin, and gentamicin with diverse mechanisms against MRSA2. Our study highlighted the significance of the allyl group in the core xanthone structure for combating MRSA2 resistance. The effectiveness of α-MG-4 in enhancing the antibacterial activity of penicillin was further supported by an MRSA2-infected skin abscess mouse model. Potential mechanisms underlying the enhancement of MRSA2 antibiotic-resistant properties by α-MG-4 were achieved by the inhibition of efflux pumps encoded by *MepA* and *NorA* and suppression of PBP2a activity via occupation of its crucial site. Taken together, α-MG-4, derived from xanthone α-MG, demonstrated a promising antimicrobial synergist that broaden approaches to address the treatment of MRSA resistance.

## MATERIALS AND METHODS

### Drugs and bacteria strains and cells

Penicillin, enrofloxacin, gentamicin, azithromycin, tetracycline, rifampicin, clindamycin, nitrocefin, polymyxin, and vancomycin were purchased from Shanghai Yuanye Biotechnology Co. Ltd. The bacterial strains utilized included *S. aureus* ATCC 29213, *E. coli* ATCC25922, MRSA2, and carbapenem-resistant *Enterobacteriaceae* (CRE-1). For cytotoxicity evaluation, human normal liver cells (L0-2) and human lung alveolar basal epithelial cells for adenocarcinoma (A549) were purchased from ATCC.

The α-MG derivatives listed in Fig. 1 were synthesized as previously described (22). The synthetic pathways were described as follows.

## α-MG-1

Under the atmosphere of $N_2$, α-MG (1 mmoL, 410 mg) was dissolved in anhydrous tetrahydrofuran (30 mL), and $LiAlH_4$ (5 mmoL, 190 mg) was added in batches at room temperature, and the reaction solution was stirred for 3 h at 70℃. When the reaction was over, and saturated ammonium chloride solution (20 mL) and ethyl acetate (50 mL) was added to the reaction mixture. The organic layer was washed with water (50 mL) and brine (50 mL) in order and dried by anhydrous $Na_2SO_4$ and filtered and then evaporated under the reduced pressure to obtain the crude product, which was purified by silica gel-based column chromatography with ethyl acetate and petroleum ether (vol/vol = 1:8) as an eluent to produce 400 mg of compound α-MG-1 as a yellow solid. The yield was 62%.

## α-MG-2, α-MG-5, and α-MG-6

To a solution of α-MG (1 mmoL, 410 mg) dissolved in anhydrous dichloromethane (20 mL) were added acetate anhydride (6 mmoL, 612 mg) and 10 drops of pyridine, and the reaction solution was stirred for 2 h at room temperature. When the reaction was done, ethyl acetate (100 mL) and water (50 mL) were added to dilute the reaction solution. The organic layer was washed with water (50 mL) and brine (50 mL) in order and dried by anhydrous $Na_2SO_4$ and filtered and then evaporated under the reduced pressure to offer the crude product, which was purified by silica gel-based column chromatography with ethyl acetate and petroleum ether (vol/vol = 1:8) as an eluent to afford 272 mg of compound α-MG-5 as a yellow solid. The yield was 55%. The synthetic methods for α-MG-2 and α-MG-6 were similar to those of α-MG-5, and only the amount of acetic anhydride was different.

## α-MG-3

α-MG (1 mmoL, 410 mg) was taken up in anhydrous DMF (15 mL), and 60% NaH (6 mmoL, 240 mg) was added in batches at 0~5℃, and benzyl bromide (6 mmoL, 942 mg) was dropped slowly to the reaction solution. The reaction was heated to 50℃ and stirred for 12 h. When the reaction was completed, ethyl acetate (100 mL) and water (50 mL) were added to dilute the reaction solution, and the operation was repeated twice. The combined organic layer was washed with water (100 mL) twice and brine (50 mL) in order and dried by anhydrous $Na_2SO_4$, and then, the filter was evaporated under the reduced pressure to obtain the crude product, which was purified by silica gel-based column chromatography with ethyl acetate and petroleum ether (vol/vol = 1:10) as an eluent to produce 531 mg of compound α-MG-3 as a yellow solid. The yield was 78%.

## α-MG-4

Under the atmosphere of $N_2$, α-MG (1 mmoL, 410 mg) and anhydrous potassium carbonate (4 mmoL, 552 mg) were taken up into anhydrous acetone (20 mL), and allyl bromide (3 mmoL, 306 mg) was added. The reaction was stirred overnight at the room temperature. When the reaction was over, the resulting solution was concentrated under the reduced pressure. Ethyl acetate (100 mL) and water (50 mL) were added to dissolve the residue. The organic layer was washed with water (50 mL) and brine (50 mL) and dried by anhydrous $Na_2SO_4$ and filtered and then concentrated under the reduced pressure to obtain the crude product, which was purified by silica gel-based column chromatography with ethyl acetate and petroleum ether (vol/vol = 1:8) as an eluent to afford 407.2 mg of compound α-MG-4 as a yellow solid. The yield was 83%.

## α-MG-7 and α-MG-8

At the atmosphere of nitrogen, to a solution of α-MG (1 mmoL, 410 mg) and 4 *N,N*-dimethyl pyridine (DMAP, 0.2 mmoL, 24.4 mg) dissolved in 20 mL of anhydrous pyridine

was added isobutyryl chloride (6 mmoL, 640 mg), and the reaction mixture was heated to 60°C and stirred overnight at the same time. When the reaction was done under the monitor of TLC (thin layer chromatography), ethyl acetate (100 mL) and water (50 mL) were added to dilute the reaction solution. The organic layer was washed with water twice (100 mL) and brine (50 mL) and dried by anhydrous $Na_2SO_4$, and then, the filter was evaporated under the reduced pressure to obtain the residue, which was purified by silica gel-based column chromatography with ethyl acetate and petroleum ether (vol/vol = 1:14) as an eluent to produce 558.6 mg of compound α-MG-7 as a yellow solid. The yield was 90%.

The synthetic procedure of α-MG-8 was similar to α-MG-7; only the amount of isobutyryl chloride was different.

## α-MG-9

Under the atmosphere of $N_2$, α-MG (1 mmoL, 410 mg) and anhydrous potassium carbonate (4 mmoL, 552 mg) were dissolved in anhydrous acetone (20 mL), and 6-bromo-1-hexanol (4 mmoL, 724 mg) was added. The reaction was stirred at 70°C. When the reaction was over under the monitor of TLC, the resulting solution was filtered and concentrated under the reduced pressure. Ethyl acetate (100 mL) and water (50 mL) were added to dissolve this residue. The organic layer was washed with water (50 mL) and brine (50 mL) in order and dried by anhydrous $Na_2SO_4$ and filtered and then evaporated under the reduced pressure to obtain the crude product, which was purified by silica gel-based column chromatography with ethyl acetate and petroleum ether (vol/vol = 1:8) as an eluent to afford 531.4 mg of compound α-MG-9 as a yellow solid.

## α-MG-10

Under the atmosphere of $N_2$, α-MG (1 mmoL, 410 mg) and anhydrous potassium carbonate (4 mmoL, 552 mg) were taken up into anhydrous acetone (20 mL), and allyl bromide (3 mmoL, 306 mg) was added. The reaction was stirred overnight at room temperature. When the reaction was over, and the resulting solution was concentrated under the reduced pressure. Ethyl acetate (100 mL) and water (50 mL) were added to dissolve the residue. The organic layer was washed with water (50 mL) and brine (50 mL) and dried by anhydrous $Na_2SO_4$ and filtered and then concentrated under the reduced pressure to obtain the crude product, which was purified by silica gel-based column chromatography with ethyl acetate and petroleum ether (vol/vol = 1:8) as an eluent to afford 407.2 mg of intermediate A as a yellow solid. The yield was 63%.

To a solution of intermediate A (1 mmoL, 490.6 mg) dissolved in anhydrous pyridine (15 mL) were added isobutyryl chloride (2 mL) and DMAP (0.2 mmoL, 24.5 mg) under the atmosphere of $N_2$. The reaction mixture was stirred at 50°C for 2 h. The resulting reaction solution was concentrated by the reduced pressure at 60°C. Ethyl acetate (50 mL) and water (30 mL) were added to dissolve the reaction mixture. The organic layer was washed with water (50 mL) and brine (50 mL) in order and dried by anhydrous $Na_2SO_4$ and filtered and then concentrated under the reduced pressure to obtain the residue, which was purified by silica gel-based column chromatography with ethyl acetate and petroleum ether (vol/vol = 1:10) as an eluent to afford 511.3 mg of compound α-MG-10 as a yellow solid.

## α-MG-11

Under the atmosphere of $N_2$, α-MG-4 (1 mmoL) and anhydrous potassium carbonate (4 mmoL, 552 mg) were taken up into anhydrous acetone (20 mL), and methyl 6-bromo-hexanoate (3 mmoL) was added. The reaction was stirred overnight at room temperature. When the reaction was over, the resulting solution was concentrated under the reduced pressure. Ethyl acetate (100 mL) and water (50 mL) were added to dissolve the residue. The organic layer was washed with water (50 mL) and brine (50 mL) and dried by anhydrous $Na_2SO_4$ and filtered and then concentrated under the reduced

pressure to obtain the crude product, which was purified by silica gel-based column chromatography with ethyl acetate and petroleum ether (vol/vol = 1:8) as an eluent to afford 407.2 mg of α-MG-10 as a yellow solid. The yield was 33%.

## Animals

Male SPF (specific pathogen free)-grade ICR (Institute of Cancer Research) (4–5weeks old, 20–25 g) mice were purchased from Beijing Speiford Biotechnology Co. Ltd., Suzhou Branch (China), the animal protocal was approved by the Committee of Experimental Animal Center of Shanghai Institute of Veterinary Medicine, China (SV-20240112-G01).

## Minimum inhibitory concentration assay and time-kill assay

The MIC was determined following the Clinical and Laboratory Standards Institute guideline (36). Briefly, the MIC of α-MG derivatives or antibiotics was determined using the checkerboard method in Mueller-Hinton broth (MHB) at an inoculum of $5 \times 10^5$ CFU/mL. The checkerboard microdilution method was further applied to evaluate the synergy between α-MG derivatives and antibiotics against various bacterial strains. Vancomycin as a control for Gram-positive bacteria and enrofloxacin as a control against Gram-negative bacteria. The MICs of each combination were monitored as described above, and subsequently, the FICI was calculated using the following formula: FICI = $MIC_{A(combination)}/MIC_A + MIC_{B(combination)}/MIC_B$. The synergetic effect was defined as an FICI of <0.5 (37).

Time-kill experiments were conducted following the procedure reported by Janardhanan et al. and Zhou Y et al. (38, 39), with minor modifications, using exponential-phase MRSA2 as a bacteria strain. Bacteria cultures were exposed to various concentrations of α-MG derivatives and antibiotics.

## Hemolytic activity and cytotoxicity

Rabbit erythrocytes were utilized to evaluate the *in vitro* hemolytic activity of the compounds with minor adjustments (40). Briefly, a solution of α-MG derivatives was added to rabbit erythrocytes, followed by incubation. The mixture was then centrifuged, and the absorbance of the supernatant was measured at 540 nm.

The cytotoxicity of the tested compounds was determined using a modified CCK-8 assay (41). A549 and L0-2 cells were employed as evaluation models. At 37°C, the two cell lines were cultivated for 24 h in 96-well microplates with an initial concentration of $1 \times 10^4$ cells/well in a humidified atmosphere with 5% $CO_2$. Then, the cultured cells were treated with different concentrations of the tested compounds followed by incubation for 24, 36, and 72 h. CCK-8 was added and incubated for another 1 h. The optical density of the dissolved material was measured at 450 nm with a spectrophotometer.

## *In vivo* mouse abscess model

The procedure for the animal infection model and assay was established based on a previous method (22). MRSA2 was cultured to the middle logarithmic growth phase, washed, and resuspended in PBS. ICR mice were anesthetized using 1% sodium barbiturate, and 100 µL of MRSA2 ($OD_{600nm} = 0.35$) was injected into the mid back of each ICR mice. In the *in vivo* experiment, the mice were divided into six groups, each group consisting of five mice (Table 3). Following the bacterial inoculation, the six groups were dosed at 1 h and 24 h. All drugs were subcutaneously administered close to the infection site. Abscesses were photographed after 24 h. On day 2, abscesses were excised and homogenized with PBS prior to serial dilution and drop plating. The abscess burden was calculated as the number of CFU. The inflammatory level of cytokine IL-6 was quantified by enzyme-linked immunosorbent assay (Sangon Biotech, Shanghai, China) following the manufacturer's instructions.

**TABLE 3** Grouping and dosing of the experimental mice ($n = 30$)[a]

| Group | Drug | Dosage |
|---|---|---|
| 1 | Control (negative control) | 100 µL solvent |
| 2 | Model | 100 µL saline |
| 3 | Vancomycin (positive control) | 2.5 mg/kg |
| 4 | α-MG-4 | 40 mg/kg |
| 5 | Penicillin | 5 mg/kg |
| 6 | α-MG-4 + penicillin | 40 mg/kg + 5 mg/kg |

[a]Solvent: Tween/PEG/ethanol/PBS = 10/35/10/45, V/V/ V.

## SEM and TEM analyses

The sample treatment and determination method were slightly modified based on the previous method (42, 43). SEM and TEM were used to observe the morphological and structural changes of bacteria after administration. For SEM observations, MRSA2 was inoculated in MHB and incubated at 37°C for 4 h and collected by centrifugation at 4,000 rpm for 10 min, and then, the pellet was washed and resuspended in the sterile PBS, and ultimately, the suspension was adjusted to $OD_{600nm} = 0.4$. The different concentrations of the drug were added to the bacterial suspension. After the incubation was performed at 37°C for 4 h, the suspension was centrifuged at 4,000 rpm for 10 min, and the supernatant was discarded and washed twice with PBS, and then, two copies of the above samples were fixed with a final concentration of 2.5% glutaraldehyde in the bacterial solution. The glutaraldehyde was washed off with PBS, and the post fixative was added and then incubated for 1.5 h. All the samples were washed twice with PBS and then dehydrated with 50%, 70%, 90%, and 100% (vol/vol) ethanol and ultimately dried and gilded using a $CO_2$ critical point dryer, and the treated samples were observed under a SEM.

The samples ($OD_{600nm} = 0.4$) used for TEM were washed twice with PBS and then dehydrated with 50%, 70%, and 90% ethanol, and then, the bacterial intracellular fluid was replaced with acetone and resin. Finally, the bacterial sample was placed in a 60°C oven for polymerization for 48 h. The processed sample slices were fixed in a copper mesh and then stained with a sodium phosphotungstate solution. After staining, the sample can be observed under a TEM and photographed for recording.

## Live/dead bacteria staining

The staining assay was modified based on the previous method for evaluating the integrity of the cell membrane (44). The operational procedure for the MRSA2 samples used for live/dead bacterial staining was similar to that of samples for SEM assays. The precipitate was mixed with DAPI and YO-PRO-1 dyes and washed at room temperature without exposure to light to obtain the stained bacterial solution, which was observed using a fluorescence microscope.

## EtBr accumulation assay

The analysis was slightly modified based on the previous method (22). MRSA2 in the logarithmic growth phase was resuspended in sterile PBS (with 0.4% glucose) and adjusted to $OD_{600nm} = 0.4$. α-MG-4 (8 mg/L) was added to the suspension. Aliquots of 0.1 mL of the above working solution were transferred into a 96-well plate. EtBr was added to each well to reach a final concentration of 5 mg/L. Microplates (BioTek) were utilized to detect fluorescence intensity, and fluorescence data were recorded every minute for 1 h at 37°C (Ex = 544 nm, Em = 590 nm).

## Intracellular enrofloxacin content assay

The analysis was slightly modified according to the previous work (45). MRSA2 solution used for an intracellular enrofloxacin content assay was similar to that of SEM assay.

MRSA2 solution was divided into three groups including a negative control group, which only contained enrofloxacin, the CCCP, and enrofloxacin-treated group, and the α-MG-4- and enrofloxacin-treated group was added with 10 MIC enrofloxacin added to each group with 5 mL of bacterial solution and cultured for 1 h. Each group was further divided into two parallels consisting of the whole liquid portion and the precipitation portion, which were incubated with lysozyme (5 mg/mL) for 0.5 h at 37°C. Subsequently, the quantitative water was added to the whole liquid group and the precipitation group, and the bacteria were lysed by freeze thawing for three times. The bacterial liquid was centrifugated at a high speed of 12,000 rpm, yielding the supernatant. After being filtered, the content of enrofloxacin was analyzed by HPLC (Agilent, C18 column: 250 × 4.6 mm × 5 µm, wavelength: 278 nm, mobile phases: water with 0.1% formic acid as phase A and acetonitrile as phase B, equal gradient elution: 86% A and 14% B). Under the conditions, the retention time of CCCP and α-MG-4 had no effect on the enrofloxacin assay. The enrofloxacin content in each group was evaluated as a ratio of enrofloxacin content in the precipitation group to that in the whole liquid group. All the experiments were performed in triplicate.

## Quantitative RT-PCR assay

The quantitative reverse transcriptase PCR (RT-PCR) assays were conducted in accordance with the previous method (46). The experimental procedure for MRSA2 samples used in quantitative RT-PCR assays was similar to that of samples for SEM assays. Total RNA was extracted using the RNAprep pure cultured cell/bacterial total RNA extraction kit (TIANGEN), followed by reverse transcription to cDNA using the PrimeScript RT kit (TaKaRa). Quantitative RT-PCR assays were performed on the Applied Biosystems 7500 Real-Time PCR System (USA). Primers for qRT-PCR (quantitative real time polymerase chain reaction) were listed in Table 4, with the 16S rRNA gene serving as the housekeeping gene.

## Western blot analysis

The western blotting assay was conducted following the previous method (39). After 4 h of treatment with different concentrations of α-MG-4, equal amounts of bacterial cultures were centrifuged, collected, mixed with 5 × loading buffer, and boiled. Each sample was separated by 10% SDS-PAGE, transferred, blocked, and incubated with anti-PBP2a primary antibodies (SAB4200853-100UL, Sigma-Aldrich) and HRP-conjugated anti-mouse secondary antibodies (1:1,000; RRID: A0216; Bitech). Finally, the membrane was visualized using an enhanced chemiluminescence.

## Bocillin FL PBP binding assay

The PBP blinding assay was slightly modified according to the previous method (47). Recombinant *S. aureus* PBP2a protein (RayBiotech) at a concentration of 1 µM in 25 mM HEPES (pH = 7) was incubated at 37°C in the presence of varying concentrations of α-MG-4 for 30 min. Subsequently, 20 µM Bocillin FL (Therm.) was added and further incubated for 30 min, followed by quenching with the addition of Laemmli sample buffer (2× stock solution) and boiling. The samples were then centrifuged and loaded onto

**TABLE 4** Sequence of primers used for RT-PCR analysis

| Gene | Primer | Sequence (5′–3′) |
| --- | --- | --- |
| 16S rRNA | Forward | CAGGTGCGAATGATGTGTGG |
| | Reverse | ACAATCCTTCCATTGGCGTG |
| *MepA* | Forward | TTATGGAAACTTCGCGATTGC |
| | Reverse | AACACCTTCACATAATCCCATGATAAT |
| *NorA* | Forward | GATTGGTGGATTTATGGCAG |
| | Reverse | GAAAAGTGCCCCAAATATACC |

10% SDS-PAGE gels, which were visualized immediately using a ChemiScope 6100 (Clinx, shanghai, China).

## β-Lactamase activity assay

Nitrocefin was used as a reaction substrate to detect the anti-β-lactamase enzymatic activity of α-MG-4 according to the previous method (48). β-Lactamase (Yuanye Bio-Technology) was dissolved in PBS (pH = 7) and stored at −20°C for further use. α-MG-4 was added to a 96-well plate that consisted of enzyme solution with a final concentration of 100 IU and Nitrocefin solution 20 µM in 25 mM HEPES buffer (pH = 7.3), and the mixture was incubated at 37°C for 1 h. The absorbance was detected by a microplate reader at 490 nm.

## Statistical analyses

All statistical analyses were performed using GraphPad Prism 8. A two-way ANOVA (analysis of variance) was employed to assess significant effects resulting from treatment with α-MG-4 and/or penicillin, enrofloxacin, and gentamicin. Statistical significance was indicated with * or #. $P$ value < 0.05, * and # ≤ 0.05–0.01, ** and ## ≤ 0.01–0.001, *** and ### ≤ 0.001–0.0001, and **** and #### ≤ 0.0001.

## ACKNOWLEDGMENTS

This work was supported by the National Key Research and Development Program of China (2023YFD1800800 and 2023YFD1801301) and National Natural Science Foundation of China (U22A20518, 32473084, and 32171931).

None of the authors have any financial or personal affiliations with individuals or organizations that could influence the outcomes of this work. R.G. performed biological assays; H.Z. collected and analyzed the data; Q.T. synthesized α-mangostin derivatives and other compounds; K.C. discussed results and data analyses and wrote and revised a manuscript; H.B. conducted the *in vivo* experiments; X.W. performed ethidium bromide accumulation assay; K.Z. verified the analytic results; W.Y. evaluated cytotoxicity assay; X.H. discussed results and aided in interpreting results; C.W. provided critical feedback and revised the manuscript; W.Z. designed and supervised the project, developed the theoretical framework, and revised the manuscript; R.J., H.Z., and D.E. wrote the manuscript in consultation with C.W., K.C. and W.Z; all authors discussed the results and contributed to the final version of the manuscript.

## AUTHOR AFFILIATIONS

[1]School of Pharmaceutical Sciences, South-Central Minzu University, Wuhan, China
[2]Shanghai Veterinary Research Institute, Chinese Academy of Agricultural Sciences, Shanghai, China
[3]Department of Pharmaceutical Chemistry, Faculty of Pharmaceutical Sciences, Prince of Songkla University, Hat Yai, Songkhla, Thailand
[4]Drug Delivery System Excellence Center, Faculty of Pharmaceutical Sciences, Prince of Songkla University, Hat-Yai, Songkhla, Thailand
[5]Department of Clinical Pharmacy, College of Pharmacy, Guilin Medical University, Guilin, Guangxi, China
[6]Key laboratory of Veterinary Chemical Drugs and Pharmaceutics, Ministry of Agriculture and Rural Affairs, Shanghai Research Institute, Chinese Academy of Agricultural Sciences, Shanghai, China

## AUTHOR ORCIDs

Keyu Zhang  http://orcid.org/0000-0002-6920-7830
Chunmei Wang  http://orcid.org/0000-0001-9860-6876
Wen Zhou  http://orcid.org/0000-0002-3961-1816

## FUNDING

| Funder | Grant(s) | Author(s) |
|---|---|---|
| MOST \| National Key Research and Development Program of China (NKPs) | 2023YFD1800800 | Wen Zhou |
| MOST \| National Key Research and Development Program of China (NKPs) | 2023YFD1801301 | Chunmei Wang |

## AUTHOR CONTRIBUTIONS

Rile Ge, Conceptualization, Data curation, Formal analysis, Investigation, Methodology, Project administration, Resources, Software, Supervision, Validation, Visualization, Writing – original draft, Writing – review and editing | Haiyan Zhao, Conceptualization, Data curation, Formal analysis, Writing – review and editing | Qun Tang, Project administration | Kasemsiri Chandarajoti, Writing – review and editing | Han Bai, Methodology | Xiaoyang Wang, Methodology | Keyu Zhang, Methodology | Wenchong Ye, Data curation | Xiangan Han, Project administration | Chunmei Wang, Formal analysis, Funding acquisition, Writing – review and editing | Wen Zhou, Funding acquisition, Methodology, Writing – review and editing

## ADDITIONAL FILES

The following material is available online.

### Open Peer Review

**PEER REVIEW HISTORY (review-history.pdf).** An accounting of the reviewer comments and feedback.

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
