## [Reviewer comments · Microbiology Spectrum]

Microbiology Spectrum

A novel α -mangostin derivative synergistic to antibiotics against MRSA with unique mechanisms

Rile Ge, Haiyan Zhao, Qun Tang, Kasemsiri Chandarajoti, Han Bai, Xiaoyang Wang, Keyu Zhang, Wenchong Ye, Chunmei Wang, Wen Zhou, and Xianghan Han

Corresponding Author(s): Chunmei Wang, Shanghai Veterinary Research Institute Chinese Academy of Agricultural Sciences

Review Timeline:

Submission Date:	July 5, 2024
Editorial Decision:	August 12, 2024
Revision Received:	September 24, 2024
Accepted:	October 8, 2024

Editor: Felix Toka

Reviewer(s): Disclosure of reviewer identity is with reference to reviewer comments included in decision letter(s). The following individuals involved in review of your submission have agreed to reveal their identity: Erdal Özbek (Reviewer #1); Hafidha Salim AL-Hattali (Reviewer #2)

Transaction Report:

DOI: <https://doi.org/10.1128/spectrum.01631-24>

Re: Spectrum01631-24 (A novel α -mangostin derivative synergistic to antibiotics for increasing the anti-MRSA activity via the inhibition of PBP2a and efflux pump)

Dear Ms. Chunmei Wang:

Thank you for the privilege of reviewing your work. Below you will find my comments, instructions from the Spectrum editorial office, and the reviewer comments.

The title and abstract could be refined, ensuring proper proofreading, clarifying methods, expanding the explanation of results, and providing a more balanced discussion by addressing conflicting findings from other studies. There is need for consistency in formatting (such as reference placement and capitalization in tables) and improvements in the content, including the addition of missing sources, and the inclusion and interpretation of statistical results in the findings and discussion sections. The specific data from the tables should be integrated into the findings section.

Revision Guidelines

Sincerely,
Felix Toka
Editor
Microbiology Spectrum

Reviewer #1 (Comments for the Author):

Dear editor,

I made some clarifications and corrections in the attached file. I have indicated the corrections with a yellow fill. In addition to these, the following must be corrected or reorganized in the article:

1. According to the journal rules, references should be written in parentheses before punctuation marks at the end of the sentence.
2. The source(s) should be added to some of the places I have specified in the additional file.
3. The last paragraph in the introduction should include the purpose of the study. However, this paragraph has mentioned the results of the study. The authors should rewrite this paragraph in line with the purpose of the study.
4. The statistical method applied is stated in the material method section. However, statistical results are not included in the findings and are not interpreted in the discussion section. Statistical results should be added and evaluated.
5. In Table 3, the synergistic effects in compounds 2 and 3 are seen to be similar to compound 4. However, the author did not mention this situation in the findings section. These data should be added to the findings section.
6. In Table S1, some of the antibiotic names start with uppercase letters and some with lowercase letters. All of them should be written the same way.

Reviewer #2 (Comments for the Author):

General feedback on the manuscript:

This study presents a promising approach to enhancing the efficacy of existing antibiotics against MRSA through the use of a novel α -mangostin derivative. The experimental design is well-structured, using both in vitro and in vivo models to assess the efficacy of the α -mangostin derivatives. The use of multiple assays supports the thoroughness of the approach. With some improvements in the presentation and critical discussion of results, this research could contribute to the field of antimicrobial therapy.

Specific Comments:

1- Title and Abstract

- The title could be concise.

- The abstract Re-write the abstract.

o The abstract should contain the following: 1-background of the topic and the main aim of the study. 2- Method and result can be combined and only the important findings are highlighted with specific statistical outcomes. 3- Conclusion needs to be more general and summarise the impact of this study.

2- Introduction

- Require proof reeding!

- "Antimicrobial sensitizers"... require a definition

- "we" could be removed

- The introduction could benefit from a previous studies involving α -mangostin and its derivatives.

- It might be useful to include an explanation of why compound 4 was selected over others.

3- Methods

- Require proof reeding!

- The synthesis process of α -MG derivatives should be briefly described or referenced more clearly.

- Instead of Compound 1, 2....4, the α -MG derivatives could be labelled as α -MG-1, α -MG-2,....., α -MG-4

- The statistical methods used to analyze the data should be explained in greater detail.

- Information on control groups and how they were handled is limited.

4- Results

- Require proof reeding!

- "compound 4" or "4" could be replaced with other term e.g α -MG-4

- Some results could be explained in more detail, such as the specific mechanisms by which compound 4 enhances membrane permeability and inhibits efflux pumps.

- The statistical significance of results should be highlighted more consistently throughout the text.

- The in vivo results could be expanded to include a discussion of potential side effects observed in the animal models.

5- Discussion:

- The manuscript could improve by discussing any conflicting findings from other studies, if any, offering a more balanced view and situating its contributions within the current scientific debate.

A novel α -mangostin derivative synergistic to antibiotics for increasing the anti-MRSA activity via the inhibition of PBP2a and efflux pump

Rile Ge^{1,2#}, Haiyan Zhao^{1#}, Qun Tang², Kasemsiri Chandarajoti^{3,4}, Han Bai⁵, Xiaoyang Wang^{2,6}, Keyu Zhang^{2,6}, Wenchong Ye^{2,6}, Xiangan Han², Chunmei Wang^{2,6*}, Wen Zhou^{2,6*}

¹School of Pharmaceutical Sciences, South-Central Minzu University, 430074, Wuhan, China

²Shanghai Veterinary Research Institute, Chinese Academy of Agricultural Sciences, 200241, Shanghai, China

³Department of Pharmaceutical Chemistry, Faculty of Pharmaceutical Sciences, Prince of Songkla University, Hat Yai, Songkhla 90112, Thailand.

⁴Drug Delivery System Excellence Center, Faculty of Pharmaceutical Sciences, Prince of Songkla University, Hat-Yai, Songkhla 90112, Thailand.

⁵ Department of Clinical Pharmacy, College of Pharmacy, Guilin Medical University, Guilin, Guangxi Zhuang Autonomous Region 541199, P.R. China

⁶Key laboratory of Veterinary Chemical Drugs and Pharmaceutics, Ministry of Agriculture and Rural Affairs, Shanghai Research Institute, Chinese Academy of Agricultural Sciences, Shanghai 200241, China

Objectives: Methicillin-resistant *Staphylococcus aureus* (MRSA) resistant to multiple antibiotics, poses significant health safety concerns. The objective of this study is to screen potential antibacterial

* Correspondent. E-mail: wangchunmei@shvri.ac.cn

* Correspondent. E-mail: zhouwen60@126.com

enhancers, assess their synergistic effects against MRSA *in vitro* and *in vivo*, and investigate the mechanisms underlying their synergy with existing antibiotics.

Methods: To broaden the search for potential sensitizers, we conducted an evaluation of the synergy between eleven α -mangostin (α -MG) derivatives and three antibiotics in order to restore sensitivity to MRSA. Changes in cell membrane structure and a mouse skin abscess model were utilized to assess the effects of the antibacterial enhancers. Furthermore, we employed RT-PCR, immunoblotting assays, and enzyme activity tests to delve into the mechanisms underlying the enhanced antibacterial effects.

Results: Compound 4 carrying an allyl group did not show direct anti-MRSA2 activity. It exhibited favorable safety profiles. Additionally, significant synergistic antibacterial effects were observed both *in vivo* and *in vitro*. The optimal synergistic ratios of compound 4 with three antibiotics were determined. The mechanisms underlying this synergy were closely associated with changes in membrane permeability and depolarization, as well as the inhibition of the *mepA* and *NorA* genes encoding efflux pumps. Furthermore, compound 4 concentration-dependently inhibited PBP2a expression by occupying its enzyme active site.

Conclusions: A novel α -MG derivative, compound 4, was identified

as a PBP2a inhibitor, demonstrating the first report of mangostin derivatives in reversing MRSA resistance to penicillin. Furthermore, compound **4** effectively eliminated resistance to three antibiotics by enhancing cell membrane permeability and inhibiting efflux pumps.

Introduction

Staphylococcus aureus (*S. aureus*) is a zoonotic pathogen. It produces various toxins like enterotoxin and leukocycline¹, which damage host cells and tissues, causing diseases such as endocarditis, bacteremia, osteomyelitis, and necrotizing fasciitis^{2,3}. Antibiotic therapy is commonly used to treat *S. aureus* infections. However, misuse and/or overuse of antibiotics have led to the rise of resistant bacteria, especially methicillin-resistant *Staphylococcus aureus* (MRSA)⁴. MRSA isolates are commonly resistant to numerous antibiotics, resulting in life-threatening infections^{5,6}. Recognizing the global health threat, the World Health Organization (WHO) considers MRSA a "high priority" pathogen⁷. Therefore, developing new strategies to fight MRSA infections is urgent.

Compared to developing new antibacterial drugs, utilizing drug combinations presents an attractive approach for controlling resistant bacterial infections. These combinations not only save time, they also reduce the doses, and cost effectiveness in treatments. Antimicrobial sensitizers, when combined with existing antibiotics, offer advantages such as reversing bacterial resistance, lowering antibiotic dosage, and

preventing the emergence of resistant strains. Plants serve as the primary source of sensitizers. A variety of plant secondary metabolites that alleviates pathogenic attacks⁸ including baicalin⁹, quercetin¹⁰, epicatechin¹¹, and ginkgo biloba flavonoids¹², demonstrating good antibacterial synergistic effects. α -Mangostin (α -MG), a xanthone compound isolated from *Garcinia cambogia*, exhibits multiple biological activities, including anti-inflammatory¹³⁻¹⁷, antibacterial¹⁸, antioxidative¹⁹, anti-cancer²⁰ properties. However, its clinical application is hindered by low water solubility and hemolytic toxicity. Structural modifications to α -MG have yielded several xanthone derivatives with potential antibacterial activity and reduced toxicity, revealing the structure-activity relationship (SAR) and core pharmacophore of α -MG. Our group has shown that modifying α -mangostin with an acetyl group at C1 can decrease toxicity and enhance antibacterial efficacy by disrupting the bacterial membrane. Despite these advancements, few xanthone-derived compounds have been explored as sensitizers in combination with existing antibiotics to combat drug-resistant bacteria.

In our study, we conducted a screening of various α -mangostin derivatives. and identified one particular derivative, labeled as compound 4, that exhibited synergistic effects with several antibiotics against MRSA. This led to the reversal of resistance in MRSA strains. Compound 4 demonstrated excellent safety profiles in our evaluations. Additionally,

we assessed the antimicrobial activity of the synergistic drugs both in vivo and in vitro. Furthermore, we delved into the mechanism underlying the synergistic antimicrobial effects of compound 4.

Materials and Methods

Drugs and bacterial strains, cells

Penicillin, enrofloxacin, gentamicin, azithromycin, tetracycline, rifampicin, clindamycin, nitrocefin, polymyxin, and vancomycin were purchased from Shanghai Yuanye Biotechnology Co., Ltd. The bacterial strains utilized included *S. aureus* ATCC 29213, *E. coli* ATCC25922, MRSA2, and carbapenem-resistant Enterobacteriaceae CRE-1. For toxicity evaluation, human normal liver cells (L0-2) and human lung alveolar basal epithelial cells for adenocarcinoma (A549) applied were purchased from ATCC.

The α -MG derivatives listed in Table S1 were synthesized according to our previous procedure²².

Animals

Male of SPF grade ICR (4-5weeks old, 20-25 g) were purchased from Beijing Speiford Biotechnology Co. Ltd. Suzhou Branch (China). This study was approved by the Committee of Experimental Animal Center of Shanghai Institute of Veterinary Medicine, China (SV-20240112-G01).

Minimum Inhibitory Concentration (MIC) assay, Synergy

evaluation and Time–kill experiments

The MIC was determined following the CLSI guideline²³. Specifically, the MIC of α -MG derivatives or antibiotics was determined using the checkerboard method in Mueller-Hinton broth (MHB) at an inoculum of 5×10^5 CFU/mL. The checkerboard microdilution method was further applied to evaluate the synergy between α -MG derivatives and antibiotics against various bacterial strains. The MICs of each combination were monitored as described above, and subsequently, the Fractional Inhibitory Concentration (FIC) index was calculated using the following formula: $FICI = \frac{MIC_{A(\text{combination})}}{MIC_A} + \frac{MIC_{B(\text{combination})}}{MIC_B}$. The synergetic effect was defined as an FIC index of <0.5 ²⁴.

Time-kill experiments were conducted following the procedure proposed by Janardhanan *et al.* and Zhou Y *et al.*, with minor modifications, using exponential-phase MRSA2^{25, 26}. Cultures were exposed to various concentrations of α -MG derivatives and antibiotics.

Hemolytic activity and cytotoxicity

Rabbit erythrocytes were utilized to evaluate the in vitro hemolytic activity of the compounds with minor adjustments²⁷. Briefly, a solution of α -MG derivatives was added to rabbit red blood cells, followed by incubation. The mixture was then centrifuged, and the absorbance of the supernatant was measured at 540 nm.

The cytotoxicity of the tested compounds was determined using a modified CCK-8 assay²⁸. A549 and L0-2 cells were employed as evaluation models.

Scanning electron microscope (SEM) assays and transmission electron microscopy (TEM) analysis

The sample treatment and determination method were slightly modified based on the previous method^{29, 30}. MRSA2 treated with the drug ($OD_{600nm} = 0.4$) was resuspended in sterile PBS for 4 hours after centrifugation, followed by washing, fixing, and detection under TEM. For SEM observations, the samples only needed to be fixed, dehydrated, dried, and gold-plated.

Live/dead bacteria staining

The staining assay was modified based on the previous method for evaluating the integrity of the cell membrane³¹. The operational procedure for the MRSA2 samples used for live/dead bacterial staining was similar to that of samples for SEM assays. The precipitate was mixed with DAPI and YO-PRO-1 dyes, respectively, and washed at room temperature without exposure to light to obtain the stained bacterial solution, which was observed using a fluorescence microscope.

***In vivo* infection evaluation**

The procedure for the animal infection model and assay was established based on the previous method. MRSA2 was cultured to the

mid-log phase, washed, and resuspended in PBS. ICR mice were anesthetized using 10% chloral hydrate, 1% Sodium barbiturate, and 100 μ L of MRSA2 ($OD_{600nm} = 0.35$) was injected into the mid-back of each ICR mice. In the experiment, the mice were divided into six groups, each consisting of 30 mice (Table 1). Following the bacterial inoculation, the six groups were administered doses at 1 hour and 24 hours, respectively. All drugs were subcutaneously administered close to the infection site. Abscesses were photographed after 24 hours. On day 2, abscesses were excised and homogenized with PBS prior to serial dilution and drop plating. The abscess burden was calculated as the number of colony-forming units (CFU). The inflammatory level of cytokine IL-6 was quantified by ELISA (Sangon Biotech, Shanghai, China) following the manufacturer's instructions.

Quantitative RT-PCR assays

The quantitative RT-PCR assays were conducted in accordance with the previous method.³⁴ The experimental procedure for MRSA2 samples used in quantitative RT-PCR assays was similar to that of samples for SEM assays. Total RNA was extracted using the RNAPrep pure cultured cell/bacterial total RNA extraction kit (TIANGEN), followed by reverse transcription to cDNA using the PrimeScript™ RT kit (TaKaRa). Quantitative RT-PCR was performed on the Applied Biosystems 7500 Real-Time PCR System (USA). Primers for qRT-PCR are listed in Table

2, with the 16S rRNA gene serving as the housekeeping gene.

Western blot analysis

The western blotting assay was conducted following the previous method. After 4 hours of treatment with different concentrations of compound 4, equal amounts of bacterial cultures were centrifuged, collected, mixed with 5×loading buffer, and boiled. Each sample was separated by 10% SDS-PAGE, transferred, blocked, and incubated with anti-PBP2a primary antibodies (SAB4200853-100UL, Sigma-Aldrich) and HRP-conjugated anti-mouse secondary antibodies (1:1000; RRID: A0216; Bitech). Finally, the membrane was visualized using enhanced chemiluminescence.

Bocillin FL PBP binding assay

The PBP binding assay was slightly modified according to the previous method.³⁶ Recombinant *S. aureus* PBP2a protein (RayBiotech) at a concentration of 1 μM in 25 mM HEPES (pH 7) was incubated at 37°C in the presence of varying concentrations of compound 4 for 30 minutes. Subsequently, 20 μM Bocillin FL (Therm.) was added and further incubated for 30 minutes, followed by quenching with the addition of Laemmli sample buffer (2 × stock solution) and boiling. The samples were then centrifuged and loaded onto 10% SDS-PAGE gels, which were visualized immediately using a ChemiScope 6100 (Clinx, shanghai, China).

Ethidium bromide accumulation assay

The analysis was slightly modified based on the previous method. MRSA2 in the logarithmic growth phase was resuspended in sterile PBS (with 0.4% glucose), and adjusted to $OD_{600nm} = 0.4$. Compound 4 (8 mg/L) was added to the suspension. Aliquots of 0.1 mL of the above working solution were transferred into a 96-well plate. Ethidium bromide (EtBr) was added to each well to reach a final concentration of 5 mg/L. Microplates (BioTek) were utilized to detect fluorescence intensity, and fluorescence data were recorded every minute for 1 hour at 37 °C (Ex=544 nm, Em=590 nm).

Statistical analyses

All statistical analyses were performed using GraphPad Prism 8. A two-way ANOVA was employed to assess significant effects resulting from treatment with compound 4 and/or penicillin, enrofloxacin, and gentamicin. Statistical significance was indicated with *. p value < 0.05, * $\leq 0.05-0.01$, ** $\leq 0.01-0.001$, *** $\leq 0.001-0.0001$, and **** ≤ 0.0001 .

Results

***The in vitro* Synergistic effects of α -MG derivatives and antibiotics on MRSA**

To investigate the potential antibacterial synergistic effects of eleven

α -MG derivatives (Figure S1) on different bacteria, we chose a range of antibiotics and tested them alongside the eleven α -MG derivatives against *S. aureus* as a representative gram-positive bacterium and *E. coli* as a Gram-negative bacterium. Then we evaluated the synergistic antibacterial effects. The eleven α -MG derivatives were synthesized following our previous synthetic procedure²².

The MICs of α -MG derivatives and antibiotics against bacteria were summarized in Table S1, while the synergistic inhibitory effects of α -MG derivatives and different antibiotics on bacteria were detailed in Table 3. Specifically, compounds **2**, **3**, and **4** exhibited synergistic effects on MRSA2 with penicillin, gentamicin, and enrofloxacin, respectively (Figure 1). To identify effective α -MG-derived synergists with favorable safety profiles, we assessed their cytotoxicity and hemolytic activity. Figure 2A illustrates that when hemolysis of rabbit red blood cells reached 80%, the concentrations of the lead α -MG compound, α -MG, and compounds **2-4** were as follows: 50 mg/L, 25 mg/L, >100 mg/L, and >100 mg/L, respectively. Comparison with α -MG against A549 cells ($IC_{50}>20$ mg/L) revealed that compounds **2** and **3** exhibited slightly stronger toxicity due to an IC_{50} of less than 20 mg/L. However, compound **4** demonstrated decreased toxicity, with an IC_{50} greater than 40 mg/L (Figure 2B and C). Similarly, compound **2** ($IC_{50} = 5.9$ mg/L) exhibited markedly greater toxicity to L0-2 cells compared to α -MG.

Notably, the IC_{50} of compound **4** against L0-2 cells was greater than 40 mg/L. These findings, consistent with previously reported hemolytic activity and cytotoxicity of α -MG, underscored the origin of toxicity stemming from α -MG itself.^{22, 38} Moreover, structural modifications significantly influenced the safety profiles of α -MG derivatives. Based on the observed cytotoxicity and hemolytic activity, compound **4**, exhibiting favorable safety profiles, was selected for further exploration of its synergistic effects on MRSA in combination with existing resistant antibiotics.

We proceeded to assess the synergistic effect of compound **4** and three antibiotics on MRSA2 (Figure 3). The fractional inhibitory concentration (FIC) values of compound **4** in combination with penicillin, gentamicin, and enrofloxacin were determined to be 0.015, 0.017, and 0.018, respectively. In contrast, α -MG did not exhibit any synergistic effect with the three antibiotics. Time growth curve and time-killing curve experiments were conducted to validate the synergistic effect. Compound **4** exhibited antibacterial activity against MRSA2 within 2 hours (Figure 4A), consistent with its MIC (>256 mg/L). MRSA2 demonstrated resistance to antibiotics including penicillin, gentamicin, and enrofloxacin. However, the addition of compound **4** overcame the resistance of the three antibiotics and demonstrated synergistic effects, resulting in a significant increase in bactericidal activity. The optimal

inhibitory concentrations of compound **4** in combination with penicillin, enrofloxacin, and gentamicin were determined to be 8 mg/L plus 1 mg/L, 8 mg/L plus 0.3125 mg/L, and 16 mg/L plus 0.3125 mg/L, respectively. These concentrations were preferentially recommended for further exploration in subsequent experiments. The results of time-killing assays (Figure 4B) confirmed the corresponding sterilization concentrations of compound **4** with the aforementioned three antibiotics, thereby guiding the selection of concentrations necessary to fall within the sterilization curve.

***In vivo* therapeutic benefits of α -MG derivative plus antibiotics**

To evaluate the sensitization of the α -MG derivative **4** to antibiotics against *in vivo* MRSA, we established a murine skin abscess model infected with MRSA2. Penicillin was selected as the tested antibiotic, and we analyzed the corresponding colony-forming unit (CFU) density *in situ*, inflammatory factors, and histopathology. On day 2, administration of either **compound 4** or penicillin alone did not lead to an improvement in symptom outcome. However, co-administration of **compound 4** and penicillin resulted in a noticeable reduction in abscess size, accompanied by a significant decrease in CFU and inflammatory factors. Histological analysis via H&E staining revealed reduced inflammatory cell infiltration and intact tissue structure in specimens treated with the combination of **compound 4** and penicillin (Figure 5D). Remarkably, compared with

vancomycin, the anti-MRSA2 effect of this combination was relatively inferior. This observation was supported by residual bacteria seen in the histological sections (Figure 5A-C).

Synergistic antibacterial mechanisms of α -MG derivatives

A broad-spectrum synergistic effect of α -MG derivative **4** on the three antibiotics with different action mechanisms against MRSA2 was observed, prompting further investigation into the synergistic mechanism. Treatment with compound **4** or penicillin (1 mg/L), enrofloxacin (0.3125 mg/L), gentamicin (0.3125 mg/L), or their combinations resulted in distinct SEM and TEM images (Figure 6). Untreated MRSA2 exhibited a complete morphology, clear cell membrane boundary, and a smooth surface. Treatment with the three antibiotics failed to induce noticeable changes in surface morphology. However, exposure to **compound 4** led to slight leakage of cellular content on the bacterial surface. When co-administered, the synergistic effect caused a significant increase in bacterial content leakage, resulting in attachment to the bacterial surface and changes in bacterial morphology, including a blurry boundary of the membrane. Moreover, the observations from TEM were highly consistent with those from SEM.

The antibacterial effect of the combined dosing was further confirmed through live/dead bacteria staining, with the fluorescence intensity of the tested compounds (Figure 7). Melittin served as the

positive control. Upon treatment with **compound 4**, some green fluorescence was observed, whereas very weak green fluorescence was detected when MRSA2 was exposed to the three antibiotics individually (Figure 7B-D). However, when MRSA2 was treated with **compound 4** in combination with antibiotics, the majority of the blue area emitted green fluorescence, indicating a stronger inhibitory effect on bacteria and the occurrence of synergistic function between **4** and antibiotics with different action mechanisms (Figure 7E).

To further investigate whether the permeability of the cell membrane was associated with the efflux pump, the efflux accumulation of ethidium bromide (EtBr) was analyzed. In the blank group, the accumulation of EtBr reached a certain extent, achieving a balance by flowing out into the extracellular environment due to the absence of an efflux pump inhibitor (Figure 8A). An increased accumulation of EtBr caused by **compound 4** was observed. However, CCCP had no influence on the accumulation of EtBr, suggesting a mode of action different from that of **compound 4**. The limited increase observed with CCCP was possibly attributed to its damaging effect on the cell membrane. As anticipated, changes in the relative expression of efflux pump genes were observed (Figure 8B). Compound 4 inhibited the relative expression of the *mepA* and *NorA* genes in a concentration-dependent manner. When enrofloxacin was administered alone, the relative expression of the *mepA* and *NorA* genes

increased approximately 28 times and 5 times, respectively. However, the relative expression of these genes decreased significantly following treatment with **compound 4** in combination with enrofloxacin. This observation further underscores the importance of **compound 4** in achieving synergistic antibacterial effects, which are mediated by the inhibition of the *mepA* and *NorA* genes encoding the efflux pump. The resistance of MRSA to penicillin is commonly attributed to the emergence of penicillin-binding protein 2a (PBP2a)³⁹. We investigated the inhibitory effects of **compound 4** on these resistant proteins. At a concentration of 62.5 mg/L, the active site of the enzyme was completely occupied by **compound 4** instead of Bocillin FL (Figure 9A). Furthermore, PBP2a expression was dose-dependently inhibited, with the inhibition plateau achieved at 8 mg/L of **compound 4**. Compared to PBP2a expression in MRSA2 suppressed by penicillin alone, a significant decrease was observed in the presence of penicillin plus **compound 4** (Figure 9B). This finding from western blot assays **provided** additional evidence to explain the synergy between penicillin and **compound 4**. Additionally, enzymatic analysis revealed no inhibitory effect of **compound 4** on β -lactamase activity (Figure 9C).

Discussion

The spread of antibiotic resistance is a major threat to public health. Antimicrobial synergists have proven to be an effective strategy in

tackling this crisis.⁴⁰ Plants, which contain a variety of functional components, serve as a primary source for reversing resistance to existing antibiotics and restoring sensitivity in resistant bacteria.⁸

In our previous research, we explored modifications to the structure of α -MG and discovered certain powerful antimicrobial derivatives. Among these derivatives (Figure 1), compound 4, which initially lacked antibacterial activity, showed the best synergy with tested antibiotics. This restored the antibacterial effectiveness of penicillin, enrofloxacin, and gentamicin against MRSA (Table 3). Notably, the safety profiles of compound 4 (Figure 2), including low hemolytic activity and minimal cytotoxicity, were superior to those of α -MG and other derivatives. The synergistic effects of compound 4 with penicillin, enrofloxacin, and gentamicin on MRSA proliferation were further confirmed through bacterial growth and bactericidal curves (Figure 4). Interestingly, in reversing the resistance of MRSA to penicillin, compound 4 demonstrated stronger elimination capability compared to the other two antibiotics. A similar synergistic effect with penicillin was observed in animal models (Figure 5), aligning well with the in vitro results (Figure 3).

It has been known that the resistance mechanisms of the three antibiotics mentioned above vary significantly. For β -lactam antibiotics like penicillin, MRSA typically employs two main resistance mechanisms. Firstly, it expresses β -lactamase, which breaks down the β -

lactam scaffold responsible for the antibiotics' antibacterial activity. Secondly, it produces PBP2a, which results in reduced affinity of β -lactam antibiotics to PBP2 TPase activity. MRSA develops resistance to aminoglycosides mainly by modifying the synthesis of aminoglycoside transferases³⁹. Resistance to fluoroquinolone antibiotics involves mutations in DNA synthase (topoisomerase II and topoisomerase IV), as well as the expression of efflux pumps that decrease drug concentration significantly⁴¹. The MFS family and the MATE family primarily control the inhibition of efflux pumps, including NorA, NorB, NorC, and mepA⁴². Additionally, changes in cell membranes are crucial targets for inhibiting bacterial resistance⁴³.

Therefore, we explored the reversal mechanisms of MRSA resistance to these three antibiotics from these various perspectives.

Since **compound 4** showed the ability to restore the susceptibility of three antibiotics, we conducted a systematic analysis of its synergistic mechanisms. We observed significant changes in the surface, morphology, and internal structure of MRSA after co-treatment with antibiotics and **compound 4** using SEM and TEM (Figure 6). These changes were consistent with previous observations indicating that α -helix peptides can increase membrane permeability, thereby exhibiting antimicrobial effects.⁴⁴ However, cleavage of cell membranes by α -helix peptide melistin was not observed in our experiments. It was

hypothesized that, similar to the membrane pores created by *Alpinia purpurata* lectin (ApuL), **compound 4** may target the cell membrane, contribute to pore formation, and consequently exert synergistic effects. Further evaluation of membrane integrity (Figure 7) supported the increased damaging ability of **compound 4** to the MRSA membrane when combined with antibiotics. This emphasizes that, similar to naturally occurring α -MG,^{46, 47} the core structure of **compound 4** enhances membrane permeability, leading to disruption of membrane integrity.

The concept of efflux pumps is widely accepted and closely linked to drug resistance. The accumulation of ethidium bromide (EtBr) serves as an indicator of efflux pump inhibition⁴⁸. Compound 4 exhibited an inhibitory effect on the efflux pump, as evidenced by increased accumulation of EtBr (Figure 8), and dose-dependent inhibition of the *mepA* and *NorA* genes. The *mepA* gene encodes the only known protein of the MATE family, which facilitates efflux of fluoroquinolones⁴². The *NorA* gene encodes the NorA protein, an efflux pump in the MFS family that can expel fluoroquinolone antibiotics⁴⁹. Bocillin FL, a fluorescent reagent, competitively occupies the active site of PBP2a, commonly used to assess inhibition activity against PBP enzymes.⁵⁰ Surprisingly, at high concentrations, **compound 4** caused the disappearance of Bocillin FL fluorescence, suggesting occupancy of the active site of PBP2a. This inhibition of PBP2a by **compound 4** occurred in a concentration-

dependent manner. No inhibition of **compound 4** was observed against β -lactamase. When administered with penicillin, co-administration of **compound 4** led to a sharp decrease in PBP2a expression and a pronounced enhancement in corresponding anti-MRSA2 activity (Figure 9). The possible mechanism is that inhibition of PBP2a by **compound 4** synergistically promotes the antibacterial activity of penicillin. It is worth noting that **compound 4**, with a xanthone structure, is the first reported PBP2a inhibitor of its kind, introducing a novel class of scaffolds for combating β -lactam antibiotic resistance.

In summary, the α -MG derivative **4**, featuring an allyl group integrated into the xanthone scaffold, exhibited a notable synergistic effect in enhancing the sensitivity of antibiotics with diverse mechanisms against MRSA. This highlights the significance of the allyl group in the xanthone structure for combating MRSA resistance. The effectiveness of **4** in enhancing the antibacterial activity of penicillin was further supported by an MRSA-infected skin abscess model. The potential mechanisms underlying the antibiotic-enhancing properties of **compound 4** include increased membrane permeability, inhibition of efflux pumps encoded by *mepA* and *NorA*, and suppression of PBP2a activity via occupancy of its active site. In conclusion, **compound 4**, derived from xanthone, represents a promising antimicrobial synergist that opens new avenues for addressing MRSA resistance.

Funding

This work was supported by National Key Research and Development Program of China (2023YFD1800802), National Natural Science Foundation of China (grant U22A20518 and 32171931), Ministry of Science and Technology of China (grant 2023YFD1800801-05 and 2023YFD1801301-05).

Transparency declarations

All authors have no financial or personal affiliations with individuals or organizations that could unduly influence our work.

Author contributions

R.G. performed biological assays. H.Z. collected and analyzed the data. Q.T. synthesized α -mangostin derivatives and other compounds. K.C. discussed results, wrote and revised the manuscript. H.B. conducted the *in vivo* experiments. X.W. performed ethidium bromide accumulation assay. K.Z. verified the analytic results. W.Y. evaluated cytotoxicity assay. X.H. discussed results and aided in interpreting results. C.W. provided critical feedback and revised the manuscript. W.Z. designed and supervised the project, developed the theoretical framework and revised the manuscript. R.J., H.Z., and D.E. wrote the manuscript in consultation with C.W., K.C. and W.Z. All authors discussed the results and contributed to the final version of the manuscript.

Supplementary data

Tables S1 and S2 are available as Supplementary data at JAC Online.

References

1. Zhang X, Hu X, Rao X. Apoptosis induced by Staphylococcus aureus toxins. *Microbiological Research* 2017; 19.
2. Tong SYC, Davis JS, Eichenberger E et al. Staphylococcus aureus Infections: Epidemiology, Pathophysiology, Clinical Manifestations, and Management. *Clinical Microbiology Reviews* 2015; **28**: 603-61.
3. Heidi W, Mansour SC, Liu LT et al. Liposomal Therapy Attenuates Dermonecrosis Induced by Community-Associated Methicillin-Resistant Staphylococcus aureus by Targeting α -Type Phenol-Soluble Modulins and α -Hemolysin. *Ebiomedicine* 2018; **33**: 211-7.
4. Jaradat Z, Ababneh Q, Shaaban S et al. Methicillin Resistant Staphylococcus aureus and public fomites: a review. *Pathogens and Global Health* 2020; **114**: 426 - 50.
5. Deng J, Zhang BZ, Chu H et al. Adenosine synthase A contributes to recurrent Staphylococcus aureus infection by dampening protective immunity. *EBioMedicine* 2021; **70**: 103505.
6. Shi M, Bai Y, Qiu Y et al. Mechanism of Synergy between Piceatannol and Ciprofloxacin against Staphylococcus aureus. *Int J Mol Sci* 2022; **23**.
7. Methicillin-resistant Staphylococcus aureus. *Nat Rev Dis Primers* 2018; **4**: 18034.
8. Shin J, Prabhakaran VS, Kim KS. The multi-faceted potential of plant-derived metabolites as antimicrobial agents against multidrug-resistant pathogens. *Microbial Pathogenesis* 2018: S0882401017317849.
9. Tang, Shusheng, Wang et al. Synergy between baicalein and penicillins against penicillinase-producing Staphylococcus aureus. *International journal of medical microbiology: IJMM* 2015; **305**: 501-4.
10. Siriwong S, Teethaisong Y, Thumanu K et al. The synergy and mode of action of quercetin plus amoxicillin against amoxicillin-resistant Staphylococcus epidermidis. *BMC pharmacology & toxicology* 2016; **17**: 39.
11. Abreu AC, McBain AJ, Simões M. Plants as sources of new antimicrobials and resistance-modifying agents. *Nat Prod Rep* 2012; **29**: 1007-21.
12. Bai Y, Wang W, Shi M et al. Novel Antibiofilm Inhibitor Ginkgetin as an Antibacterial Synergist against Escherichia coli. *Int J Mol Sci* 2022; **23**.

13. Chen WG, Zhang SS, Pan S et al. α -Mangostin Treats Early-Stage Adjuvant-Induced Arthritis of Rat by Regulating the CAP-SIRT1 Pathway in Macrophages. *Drug Des Devel Ther* 2022; **16**: 509-20.
14. Wang DD, Li Y, Wu YJ et al. Xanthenes from *Securidaca inappendiculata* antagonized the anti-rheumatic effects of methotrexate in vivo by promoting its secretion into urine. *Expert Opinion on Drug Metabolism & Toxicology* 2020.
15. Wu YJ, Zhang SS, Yin Q et al. α -Mangostin Inhibited M1 Polarization of Macrophages/Monocytes in Antigen-Induced Arthritis Mice by Up-Regulating Silent Information Regulator 1 and Peroxisome Proliferators-Activated Receptor γ Simultaneously. *Drug Des Devel Ther* 2023; **17**: 563-77.
16. Zhou Y, Xiang R, Qin G et al. Xanthenes from *Securidaca inappendiculata* Hassk. attenuate collagen-induced arthritis in rats by inhibiting the nicotinamide phosphoribosyltransferase/glycolysis pathway and macrophage polarization. *Int Immunopharmacol* 2022; **111**: 109137.
17. Yin Q, Wu YJ, Pan S et al. Activation of Cholinergic Anti-Inflammatory Pathway in Peripheral Immune Cells Involved in Therapeutic Actions of α -Mangostin on Collagen-Induced Arthritis in Rats. *Drug Des Devel Ther* 2020; **14**: 1983-93.
18. Laboratory evaluation of the antibacterial and cytotoxic effect of alpha-mangostin when used as a root canal irrigant. *Indian Journal of Dentistry* 2013; **4**: 12-7.
19. Ruankham W, Suwanjang W, Phopin K et al. Modulatory Effects of Alpha-Mangostin Mediated by SIRT1/3-FOXO3a Pathway in Oxidative Stress-Induced Neuronal Cells. *Front Nutr* 2021; **8**: 714463.
20. Hyun-Ho K, In-Ryoung K, Hye-Jin K et al. α -Mangostin Induces Apoptosis and Cell Cycle Arrest in Oral Squamous Cell Carcinoma Cell. *Evid Based Complement Alternat Med* 2016; **2016**: 5352412.
21. Sultan OS, Kantilal HK, Khoo SP et al. The Potential of α -Mangostin from *Garcinia mangostana* as an Effective Antimicrobial Agent-A Systematic Review and Meta-Analysis. *Antibiotics (Basel)* 2022; **11**.
22. Lu Y, Guan T, Wang S et al. Novel xanthone antibacterials: Semi-synthesis, biological evaluation, and the action mechanisms. *Bioorg Med Chem* 2023; **83**: 117232.
23. Wayne PA. CLINICAL AND LABORATORY STANDARDS INSTITUTE. PERFORMANCE STANDARDS FOR ANTIMICROBIAL SUSCEPTIBILITY TESTING. 2011.
24. Odds FC. Synergy, antagonism, and what the checkerboard puts between them. *Journal of Antimicrobial Chemotherapy* 2003: 1.
25. Janardhanan J, Bouley R, Martínez-Caballero S et al. The quinazolinone allosteric inhibitor of PBP2a synergizes with piperacillin and tazobactam against

methicillin-resistant *Staphylococcus aureus*. *Antimicrobial Agents & Chemotherapy* 2019.

26. Xu L, Zhou Y, Niu S et al. A novel inhibitor of monooxygenase reversed the activity of tetracyclines against tet(X3)/tet(X4)-positive bacteria. *EBioMedicine* 2022; **78**: 103943.

27. Wu TK, Wang YK, Chen YC et al. Identification of a *Vibrio furnissii* Oligopeptide Permease and Characterization of Its In Vitro Hemolytic Activity. *Journal of Bacteriology* 2007; **189**: 8215.

28. Nalbantsoy A, Nesil T, Erden S et al. Adjuvant effects of *Astragalus* saponins macrophyllsaponin B and astragaloside VII. *J Ethnopharmacol* 2011; **134**: 897-903.

29. Lu CH, Shiau CW, Chang YC et al. SC5005 dissipates the membrane potential to kill *Staphylococcus aureus* persisters without detectable resistance. *The Journal of antimicrobial chemotherapy*: dkab114.

30. Zhou L, Lian K, Wang M et al. The antimicrobial effect of a novel peptide LL-1 on *Escherichia coli* by increasing membrane permeability. *BMC Microbiology* 2022; **22**: 1-10.

31. Teng P, Nimmagadda A, Su M et al. Novel bis-cyclic guanidines as potent membrane-active antibacterial agents with therapeutic potential. *Chemical Communications* 2017; **53**.

32. Gunasekaran P, Fan M, Kim EY et al. Amphiphilic Triazine Polymer Derivatives as Antibacterial And Anti-atopic Agents in Mice Model. *Scientific Reports*.

33. Diclofenac Resensitizes Methicillin-Resistant *Staphylococcus aureus* to β -Lactams and Prevents Implant Infections. *Advanced Science* 2021.

34. Abdelraheem WM, Khairy RMM, Zaki AI et al. Effect of ZnO nanoparticles on methicillin, vancomycin, linezolid resistance and biofilm formation in *Staphylococcus aureus* isolates. *Annals of clinical microbiology and antimicrobials*; **20**: 54.

35. Lv Q, Li S, Wei H et al. Identification of the natural product paeonol derived from peony bark as an inhibitor of the *Salmonella enterica* serovar Typhimurium type III secretion system. *Applied Microbiology and Biotechnology* 2020; **104**: 1673-82.

36. Dave K, Palzkill T, Pratt RF. Neutral β -Lactams Inactivate High Molecular Mass Penicillin-Binding Proteins of Class B1, Including PBP2a of MRSA. *ACS Medicinal Chemistry Letters* 2014; **5**: 154-7.

37. Rodrigues L, Wagner D, Viveiros M et al. Thioridazine and chlorpromazine inhibition of ethidium bromide efflux in *Mycobacterium avium* and *Mycobacterium smegmatis*. *J Antimicrob Chemother* 2008; **61**: 1076-82.

38. Felix L, Mishra B, Khader R et al. In Vitro and In Vivo Bactericidal and Antibiofilm Efficacy of Alpha Mangostin Against *Staphylococcus aureus* Persister

- Cells. *Front Cell Infect Microbiol* 2022; **12**: 898794.
39. Lade H, Kim JS. Molecular Determinants of β -Lactam Resistance in Methicillin-Resistant *Staphylococcus aureus* (MRSA): An Updated Review. *Antibiotics (Basel)* 2023; **12**.
40. Wright, Gerard, D. Antibiotic Adjuvants: Rescuing Antibiotics from Resistance (vol 24, pg 862, 2016). *Trends in Microbiology* 2016.
41. Mlynarczyk-Bonikowska B, Kowalewski C, Krolak-Ulinska A et al. Molecular Mechanisms of Drug Resistance in *Staphylococcus aureus*. *Int J Mol Sci* 2022; **23**.
42. Dashtbani-Roozbehani A, Brown MH. Efflux Pump Mediated Antimicrobial Resistance by *Staphylococci* in Health-Related Environments: Challenges and the Quest for Inhibition. *Antibiotics* 2021; **10**.
43. Kim W, Hendricks GL, Tori K et al. Strategies against methicillin-resistant *Staphylococcus aureus* persisters. *Future Med Chem* 2018; **10**: 779-94.
44. Wang J, Chou S, Xu L et al. High specific selectivity and Membrane-Active Mechanism of the synthetic centrosymmetric α -helical peptides with Gly-Gly pairs. *Scientific Reports* 2015; **5**: 15963.
45. Ferreira GRS, Brito JS, Procópio TF et al. Antimicrobial potential of *Alpinia purpurata* lectin (ApuL): Growth inhibitory action, synergistic effects in combination with antibiotics, and antibiofilm activity. *Microb Pathog* 2018; **124**: 152-62.
46. Wijesundara NM, Lee SF, Cheng Z et al. Bactericidal Activity of Carvacrol against *Streptococcus pyogenes* Involves Alteration of Membrane Fluidity and Integrity through Interaction with Membrane Phospholipids. *Pharmaceutics* 2022; **14**.
47. Roshan N, Riley TV, Knight DR et al. Natural products show diverse mechanisms of action against *Clostridium difficile*. *J Appl Microbiol* 2019; **126**: 468-79.
48. Martins M, Viveiros M, Couto I et al. Identification of efflux pump-mediated multidrug-resistant bacteria by the ethidium bromide-agar cartwheel method. *Vivo* 2011; **25**: 171.
49. Costa SS, Sobkowiak B, Parreira R et al. Genetic Diversity of *norA*, Coding for a Main Efflux Pump of *Staphylococcus aureus*. *Frontiers in Genetics* 2019; **9**.
50. Zhao G. BOCILLIN FL, a Sensitive and Commercially Available Reagent for Detection of Penicillin-Binding Proteins. *Antimicrobial Agents & Chemotherapy* 1999; **43**: 1124.

General feedback on the manuscript:

This study presents a promising approach to enhancing the efficacy of existing antibiotics against MRSA through the use of a novel α -mangostin derivative. The experimental design is well-structured, using both in vitro and in vivo models to assess the efficacy of the α -mangostin derivatives. The use of multiple assays supports the thoroughness of the approach. With some improvements in the presentation and critical discussion of results, this research could contribute to the field of antimicrobial therapy.

Specific Comments:

1- Title and Abstract

- The title could be concise.
- The abstract Re-write the abstract.
 - The abstract should contain the following: 1-background of the topic and the main aim of the study. 2- Method and result can be combined and only the important findings are highlighted with specific statistical outcomes. 3- Conclusion needs to be more general and summarise the impact of this study.

2- Introduction

- Require proof reeding!
- "Antimicrobial sensitizers" ... require a definition
- "we" could be removed
- The introduction could benefit from a previous studies involving α -mangostin and its derivatives.
- It might be useful to include an explanation of why compound 4 was selected over others.

3- Methods

- Require proof reeding!
- The synthesis process of α -MG derivatives should be briefly described or referenced more clearly.
- Instead of Compound 1, 2....4, the α -MG derivatives could be labelled as α -MG-1, α -MG-2,....., α -MG-4
- The statistical methods used to analyze the data should be explained in greater detail.
- Information on control groups and how they were handled is limited.

4- Results

- Require proof reeding!
- "compound 4" or "4" could be replaced with other term e.g α -MG-4
- Some results could be explained in more detail, such as the specific mechanisms by which compound 4 enhances membrane permeability and inhibits efflux pumps.
- The statistical significance of results should be highlighted more consistently throughout the text.
- The in vivo results could be expanded to include a discussion of potential side effects observed in the animal models.

5- Discussion:

- The manuscript could improve by discussing any conflicting findings from other studies, if any, offering a more balanced view and situating its contributions within the current scientific debate.

Dear Editor and Reviewers,

We are thankful that you offered us the opportunity to revise our manuscript (Spectrum01631-24). We have revised and addressed the issues on our manuscript. In brief, we addressed the uncleared paragraph and rewrote every sections. We have resubmitted our revised manuscript and all relevant files to *Microbiology Spectrum*. If the revised manuscript still has rooms for improvement, please kindly allow us to revise it again. The following responses were addressed point-by-point.

Reviewer #1 (Comments for the Author):

Dear editor:

I made some clarifications and corrections in the attached file. I have indicated the corrections with a yellow fill. In addition to these, the following must be corrected or reorganized in the article:

1. According to the journal rules, references should be written in parentheses before punctuation marks at the end of the sentence.

Response: Thank you very much for your suggestion. We have inserted all the references carefully following the journal guideline.

2. The source(s) should be added to some of the places I have specified in the additional file.

Response: Thank you very much for your kindly help. We have added the sources within the method section in the revised manuscript.

3. The last paragraph in the introduction should include the purpose of the study. However, this paragraph has mentioned the results of the study. The authors should rewrite this paragraph in line with the purpose of the study.

Response: Thank you very much for your suggestion. We have rewritten the last paragraph of the introduction section as follows; "In this study, we synthesized and conducted a screening of various structures of α -MG derivatives against Gram-positive and Gram-negative bacteria. The results showed that α -MG substituted with an allyl at C3, affording α -MG-4, lessened toxicity to A549 and L0-2 cells and lowered the hemolysis. α -MG derivatives enhanced anti-bacterial activity by disrupting the bacterial membrane. Moreover, the selected xanthone-derived compounds were investigated as anti-MRSA2 agents in combination with various group of the existing antibiotics. We were able to identify an α -MG synergist (α -MG-4) which exhibited synergistic effects with penicillin, enrofloxacin and gentamycin against MRSA2. We investigated the synergistic effects involved in cellular mechanisms featuring bacterial membrane integrity and an efflux pump."

4. The statistical method applied is stated in the material method section. However, statistical results are not included in the findings and are not interpreted in the discussion section. Statistical results should be added and evaluated.

Response: We have addressed statistical significances for of all experiments corresponding to the

symbols (* and #) indicated at the figures.

5. In Table 3, the synergistic effects in compounds 2 and 3 are seen to be similar to compound 4. However, the author did not mention this situation in the findings section. These data should be added to the findings section.

Response: Thank you very much for pointing out about this. We have discussed about the selection of α -MG-4 at the first paragraph of the results. Although, the MIC and FICI of a α -MG-2, α -MG-3 and α -MG-4 were similar, the toxicity evaluated on rabbit red blood cells and cytotoxicity on cell lines revealed that α -MG-4 was biologically safer than α -MG-2 and α -MG-3 due to its main chemically substituents at R₂.

6. In Table S1, some of the antibiotic names start with uppercase letters and some with lowercase letters. All of them should be written the same way.

Response: Thank you very much. We have already revised the antibiotic names starting with uppercase letters across all files.

Reviewer #2 (Comments for the Author):

General feedback on the manuscript:

This study presents a promising approach to enhancing the efficacy of existing antibiotics against MRSA through the use of a novel α -mangostin derivative. The experimental design is well-structured, using both in vitro and in vivo models to assess the efficacy of the α -mangostin derivatives. The use of multiple assays supports the thoroughness of the approach. With some improvements in the presentation and critical discussion of results, this research could contribute to the field of antimicrobial therapy.

Specific Comments:

1- Title and Abstract

- The title could be concise

Response: Thank you very much for your suggestion. We have revised the title to “A novel α -mangostin derivative synergistic to antibiotics against MRSA with unique mechanisms”

- The abstract Re-write the abstract.

2- The abstract should contain the following: 1-background of the topic and the main aim of the study. 2- Method and result can be combined and only the important findings are highlighted with specific statistical outcomes. 3- Conclusion needs to be more general and summarise the impact of this study.

Response: Thank you very much for your suggestion. We have revised the two abstracts as follows:

ABSTRACT

Methicillin-resistant Staphylococcus aureus (MRSA) remains a leading cause of hospital-acquired infections, often linked to complicated treatments, increased mortality risk, and significant cost burdens. Several antibacterial agents have been developed to address MRSA resistance. In this study, potential agents to combat MRSA resistance were explored, with the antibacterial activity of synthesized α -mangostin (α -MG) derivatives being evaluated alongside

investigations into their cellular mechanisms against MRSA2. α -MG-4, featuring an allyl group at C3 of the lead compound α -MG, restored the sensitivity of MRSA2 to penicillin, enrofloxacin, and gentamicin, while also demonstrating improved safety profiles. Although α -MG-4 alone was ineffective against MRSA2, it exhibited an optimal synergistic ratio in vitro when combined with these antibiotics. This significant synergistic antibacterial effect was further confirmed in vivo using a mouse skin abscess model. Additionally, the synergistic mechanisms revealed that α -MG-4 was associated with changes in membrane permeability and inhibition of the *mepA* and *NorA* genes, which encode the efflux pumps of MRSA2. α -MG-4 also inhibited PBP2a expression, potentially by occupying a crucial binding site in a dose-dependent manner.”

IMPORTANCE

MRSA's resistance to multiple antibiotics poses significant health and safety concerns. A novel α -mangostin derivative, α -MG-4, was first identified as a xanthone-based PBP2a inhibitor that reverses MRSA2 resistance to penicillin. The synergistic antibacterial effects of α -MG-4 were linked to increased cell membrane permeability and the inhibition of genes involved in efflux pump function.

2- Introduction

- Require proof reeding!

Response: Thank you very much. We have rewritten the introduction to be clearer and more concise.

- "Antimicrobial sensitizers"... require a definition
- "we" could be removed

Response: Thank you very much for your suggestion. We have paraphrased to remove all the words “Antimicrobial sensitizers” and “we” in the revised manuscript.

- The introduction could benefit from a previous studies involving α -mangostin and its derivatives.

Response: Thank you very much for your suggestion. We have added the information of α -MG studied by our group as follow; “Structural modifications to α -MG resulted in several xanthone derivatives with potential antibacterial activity and toxicity reduction, suggesting the importance of a core pharmacophore of α -MG and its structure-activity relationship (SAR) (1). Our previous study indicated that modifying α -MG structure with an acetyl group at C1 decreased its toxicity and enhances anti-bacterial efficacy by disrupting the bacterial membrane (2).”

- It might be useful to include an explanation of why compound 4 was selected over others.

Response: We appreciate your suggestions. We have defined “Antimicrobial sensitizers” in the introduction, and explained the reasons why α -mangostin and its derivatives were chosen for sensitizer screening. Finally, the advantages of compound 4 (α -MG-4 in the revised manuscript) over others were discussed in the results and discussion sections.

3- Methods

- Require proof reeding!

Response: Thank you very much for your suggestion. Methods were intensively revised. The

information in each assay was provided in detail.

- The synthesis process of α -MG derivatives should be briefly described or referenced more clearly.

Response: Thank you very much for your suggestion. The synthetic process of α -MG derivatives was added in Supplementary materials.

- Instead of Compound 1, 2....4, the α -MG derivatives could be labelled as α -MG-1, α -MG-2,....., α -MG-4

Response: Thank you very much for your suggestion. We have revised the compound names across all files. The compounds were renamed as α -MG-1, α -MG-2,....., α -MG-11.

- The statistical methods used to analyze the data should be explained in greater details.

Response: Thank you very much for your suggestion. The statistical methods used to analyze the data were addressed and described in details at the method section. Moreover, all methods were described more in detail. Related references were cited in the materials and methods section.

- Information on control groups and how they were handled is limited.

Response: The information of control groups was provided in detail in the revised manuscript.

4- Result

- Require proof reeding!

Response: Thank you very much. We have rewritten the results and separated results from each assay corresponding to each figure and described the results in more detail.

- "compound 4" or "4" could be replaced with other term e.g α -MG-4

Response: We have replaced "compound 4" with " α -MG-4" across all files (Tables, Figures, and Supplementary files).

- Some results could be explained in more detail, such as the specific mechanisms by which compound 4 enhances membrane permeability and inhibits efflux pumps.

Response: Thank you very much for your suggestion. We have discussed the potential mechanisms of α -MG-4 at the discussion section.

- (1) The enhanced membrane permeability was by its chemical structure containing a xanthone. Possible mechanism was that the lipotropic effect of compound containing xanthone may contribute to the pore formation at the cell membrane (3,4). Propenyl, an allyl substitution at R₂ of α -MG-4 may involve in reducing MRSA membrane integrity.
- (2) α -MG-4 exhibited an inhibitory effect on the efflux pump, as evidenced by the increased accumulation of EtBr as well as the inhibition of *MepA* and *NorA* genes in a dose-dependent manner (Figure 8).
- (3) α -MG-4 acted as a PBP2a inhibitor and not β -lactamase. PBP2a was diminished in a presence of α -MG-4 (Figure 9).

- The statistical significance of results should be highlighted more consistently throughout the

text.

Response: We have addressed statistical significances for of all experiments corresponding to the symbols (* and #) indicated at the figures.

- The *in vivo* results could be expanded to include a discussion of potential side effects observed in the animal models.

Response: Thank you very much. We have discussed our observation regarding the side effects at discussion section. We did not observe the discomforts resulted from the abscesses in mice. There were no physical and apparent symptoms presented related to the potential side effects such as skin irritation or skin rash or skin inflammation at the site of the SC injection. We did not observe skin reaction in the mice treated with α -MG-4 (40 mg/mL) alone or mice treated with α -MG-4 (40 mg/mL) and penicillin (5 mg/mL). After 72 hours, mice were euthanized.

Discussion:

- The manuscript could improve by discussing any conflicting findings from other studies, if any, offering a more balanced view and situating its contributions within the current scientific debate.

Response: Thank you very much for your suggestion. We have discussed more on the chemistry view point regarding the modified structures contributed to the decreased toxicity and enhanced effects on the bacteria. In addition, we have added information regarding cellular mechanisms of the resistances and how the α -MG-4 may possibly involve. The findings that we have added were placed in the results and mainly at the discussion.

References

1. Sultan, O. S., Kantilal, H. K., Khoo, S. P., *et al.*, 2022. The Potential of α -Mangostin from *Garcinia mangostana* as an Effective Antimicrobial Agent-A Systematic Review and Meta-Analysis. *Antibiotics* (Basel, Switzerland) 11(6): 717.
2. Lu, Y., Guan, T., Wang, S., *et al.*, 2023. Novel xanthone antibacterials: Semi-synthesis, biological evaluation, and the action mechanisms. *Bioorganic & medicinal chemistry* 83: 117232.
3. Wijesundara, N. M., Lee, S. F., Cheng, Z., *et al.*, 2022. Bactericidal Activity of Carvacrol against *Streptococcus pyogenes* Involves Alteration of Membrane Fluidity and Integrity through Interaction with Membrane Phospholipids. *Pharmaceutics* 14(10): 1992.
4. Roshan, N., Riley, T. V., Knight, *et al.*, 2019. Natural products show diverse mechanisms of action against *Clostridium difficile*. *Journal of applied microbiology* 126(2): 468–479.

Re: Spectrum01631-24R1 (A novel α -mangostin derivative synergistic to antibiotics against MRSA with unique mechanisms)

Dear Dr. Chunmei Wang:

Your manuscript has been accepted, and I am forwarding it to the ASM production staff for publication. Your paper will first be checked to make sure all elements meet the technical requirements. ASM staff will contact you if anything needs to be revised before copyediting and production can begin. Otherwise, you will be notified when your proofs are ready to be viewed.

Sincerely,
Felix Toka
Editor
Microbiology Spectrum

Reviewer #1 (Comments for the Author):

Dear authors
Thank you for the revisions you have made.

Reviewer #2 (Comments for the Author):

I have reviewed the author's responses to the comments provided during the review process. I am satisfied with the responses since that all comments were effectively addressed and detailed explanations were provided. That enhanced the clarity of the manuscript.